# Integrating ecosystem markets to co-ordinate landscape-scale public benefits from nature

**Mark S. Reed**[1,2]\*, **Tom Curtis**[3], **Arjan Gosal**[4], **Helen Kendall**[1], **Sarah Pyndt Andersen**[4], **Guy Ziv**[5], **Anais Attlee**[6], **Richard G. Fitton**[7], **Matthew Hay**[8], **Alicia C. Gibson**[7], **Alex C. Hume**[7], **David Hill**[9], **Jamie L. Mansfield**[7], **Simone Martino**[10], **Asger Strange Olesen**[11], **Stephen Prior**[10], **Christopher Rodgers**[12], **Hannah Rudman**[1], **Franziska Tanneberger**[13]

**1** Department of Rural Economies, Environment & Society, Thriving Natural Capital Challenge Centre, Scotland's Rural College (SRUC), Edinburgh, United Kingdom, **2** Centre for Rural Economy and Institute for Agri-Food Research and Innovation, School of Natural and Environmental Sciences, Newcastle University, Newcastle upon Tyne, United Kingdom, **3** 3Keel LLP, Fenlock Court, Long Hanborough, United Kingdom, **4** Institute for European Environmental Policy, Bruxelles, Belgium, **5** School of Geography, University of Leeds, Leeds, West Yorkshire, United Kingdom, **6** Collingwood Investments Ltd and Project Maya CIC, London, United Kingdom, **7** FinanceEarth, London, United Kingdom, **8** Forest Carbon Ltd, Abbey Road Business Centre, Durham, United Kingdom, **9** Environment Bank Ltd, The Catalyst, York, North Yorkshire, United Kingdom, **10** Department of Environment and Geography, University of York, Heslington, York, United Kingdom, **11** Forest Stewardship Council, Freiburg, Germany, **12** School of Law, Newcastle University, 21–24 Windsor Terrace Law School, Newcastle University, Newcastle upon Tyne, United Kingdom, **13** Greifswald University, Partner in the Greifswald Mire Centre, Greifswald, Germany

\* mark.reed@sruc.ac.uk

**Data Availability Statement:** UK Data Archive https://reshare.ukdataservice.ac.uk/855274/.

**Funding:** This research was funded by the Global Food Security's 'Resilience of the UK Food System

## Abstract

Ecosystem markets are proliferating around the world in response to increasing demand for climate change mitigation and provision of other public goods. However, this may lead to perverse outcomes, for example where public funding crowds out private investment or different schemes create trade-offs between the ecosystem services they each target. The integration of ecosystem markets could address some of these issues but to date there have been few attempts to do this, and there is limited understanding of either the opportunities or barriers to such integration. This paper reports on a comparative analysis of eleven ecosystem markets in operation or close to market in Europe, based on qualitative analysis of 25 interviews, scheme documentation and two focus groups. Our results indicate three distinct types of markets operating from the regional to national scale, with different modes of operation, funding and outcomes: regional ecosystem markets, national carbon markets and green finance. The typology provides new insights into the operation of ecosystem markets in practice, which may challenge traditionally held notions of Payment for Ecosystem Services. Regional ecosystem markets, in particular, represent a departure from traditional models, by using a risk-based funding model and aggregating both supply and demand to overcome issues of free-riding, ecosystem service trade-offs and land manager engagement. Central to all types of market were trusted intermediaries, brokers and platforms to aggregate supply and demand, build trust and lower transaction costs. The paper outlines six options for blending public and private funding for the provision of ecosystem services and proposes a framework for integrating national carbon markets and green finance with regional ecosystem markets. Such integration may significantly increase funding for

Programme' with support from BBSRC, ESRC, NERC and Scottish Government, as part of the Resilient Dairy Landscapes project (grant BB/R005680/1), the Integrated Catchment Solutions Programme (iCASP) funded by the UK Natural Environment Research Council's Regional Impact from Science of the Environment scheme (grant NE/P011160/1), European Commission's Directorate-General for Climate Action (contract 340201/2018/789608/ETU/CLIMA.C.3), NatureScot (Project No. 117832) and the British Academy (grant KF5210311). Salary support was provided by 3Keel to TC, Forest Carbon Ltd to SP, Environment Bank Ltd to DH, Institute for European Environmental Policy to AG and SPA, Collingwood Investments Ltd to SA, Project Maya CIC to SA, Forest Stewardship Council to AO, Finance Earth to RF. The funders had no role in study design, data collection and analysis, decision to publish, or preparation of the manuscript.

**Competing interests:** MR is Research Lead for IUCN UK Peatland Programme and sits on the Executive Board for the Peatland Code. TC is a founding partner of 3Keel and helped develop Landscape Enterprise Networks. MH is a Project Manager and SP is co-founder of Forest Carbon Ltd. DH is founding owner of Environment Bank Ltd. AG and SPA are employees of the Institute for European Environmental Policy. SA is an employee of Collingwood Investments Ltd and Project Maya CIC. AO is an employee of Forest Stewardship Council. RF is an employee of Finance Earth. Authors' commercial affiliations do not alter our adherence to PLOS ONE policies on sharing data and materials. There are no patents, products in development or marketed products associated with this research to declare.

regenerative agriculture and conservation across multiple habitats and services, whilst addressing issues of additionality and ecosystem service trade-offs between multiple schemes.

## 1 Introduction

Worldwide, benefits from nature to society have been estimated to be worth more than the global gross domestic product [1]. When ecosystems become degraded, the cost of restoration can be prohibitive, and businesses and communities who rely most directly on these services are typically the first to suffer the consequences [2]. Neoclassical economics suggests that if property rights are clear and well defined (and if transaction costs are not too high), a social optimum can be attained via bargaining amongst ecosystem service providers and beneficiaries [3]. This sets the basis for the market to theoretically protect and sustain those services [4, 5]. While this may work for some provisioning services over short time-horizons (e.g. food and fibre), markets often fail to reward those responsible for providing service (e.g. upstream farmers or forest managers whose work benefits those downstream) when benefits are hard to attribute a financial value to (e.g. mental health or spiritual benefits from nature) or when benefits mainly accrue to others in society (e.g. downstream flood protection) over longer time-horizons (e.g. climate change mitigation). As a result, many resource management decisions generate short-term private benefits to the owner or manager at the expense of longer-term public benefits, often leading to negative externalities (e.g. pollution or flooding).

In response to this, governments commonly pay resource managers to adopt more sustainable practices and carry out other work that can protect or enhance public benefits from nature. Businesses may also pay for these public benefits for a variety of reasons, including the need to mitigate risks to their business (e.g. from climate change), reduce costs (e.g. by delivering cleaner water), secure social licence to operate or contribute towards corporate sustainability goals [1, 6–8]. This is increasingly being done via Payment for Ecosystem Service (PES) schemes, which offer monetary incentives to individuals or organisations to adopt or alter behaviours, beyond what is legally mandated, to improve the provision of ecosystem services that would otherwise have been economically unviable to provide [9–12].

However, there are a number practical challenges to the development and operation of PES schemes [13, 14]. Challenges that may deter buyers (such as food processors and water companies) and investors in ecosystem services (such as insurance companies and impact investors) include: the complexity of demonstrating the additionality and permanence of benefits (i.e. proving that they would not have happened without investment and the benefits will be long-term), costs of monitoring and verifying benefits, coordination between investors to avoid non-paying beneficiaries piggybacking on investments (i.e. benefiting from the investment of competitors without contributing themselves) or benefits for one investor cancelling out benefits for others (for example, tree planting creating habitat for predators of a species being protected by a neighbouring scheme) [15–19].

There are also many potential barriers discouraging resource managers (for example, land-owners, tenants and other businesses managing natural resources; the typical 'suppliers' whose actions shape ecosystem service delivery) from engaging in schemes. These include: poorly defined property rights, perceived (and real) risks of entering long-term contracts (including unknown impacts that managing for ecosystem services would have on land value), lack of clarity as to their eligibility for funding from public schemes after entering a privately funded (by private enterprise or investment) scheme, as well as more straightforward capacity issues relating to how they would implement and manage such schemes [3, 13, 20–24].

There is also potential for private ecosystem markets to compete with publicly funded agri-environment schemes, which are becoming increasingly PES-like in their design. For example, the latest Rural Development Programmes under the EU's Common Agricultural Policy pay more for environmental outcomes than ever before [25, 26] and post-Brexit agricultural policies in the UK are increasingly focusing on "public money for public goods" [27]. Even if publicly funded schemes pay lower amounts over shorter time-horizons than privately funded schemes, they may still displace private funding if they are perceived to be simpler or more familiar, and hence lower risk to resource managers [23].

The integration of different private ecosystem markets could address some of these issues by actively managing synergies and trade-offs. However, to date there have been few attempts to do this, and there is limited understanding of either the opportunities or barriers to integration of private markets. There is also limited analysis of interactions between public and private schemes, or how these might be better "blended". While much has been written about international voluntary and compliance carbon markets in recent years [28–30], much less is known about the national and sub-national ecosystem markets that have proliferated in recent years, and how they operate or interact with each other.

This paper therefore uses a comparative analysis of existing private ecosystem markets in operation or close to market at national and sub-national scales in the UK and elsewhere in Europe, to explore governance issues associated with integrating different types of ecosystem markets. Specifically, it aims to:

- Develop a typology of ecosystem markets by comparing ecosystem markets currently in operation or close to market in the UK, Germany, Switzerland and the Netherlands;

- Question some of the operating assumptions of ecosystem markets and offer insights that could enable the cost-effective operation of schemes that minimise trade-offs and integrate benefits across a wide range of land uses at landscape scales; and

- Propose an approach that could be used to integrate multiple ecosystem markets, operating over multiple land uses and habitats, including the integration of private markets and the blending of public and private schemes designed to deliver public goods.

The analysis includes all known private schemes operating or close to market in the UK, where the development of ecosystem markets has been a policy priority since the launch of the Woodland Carbon Code in 2011 and the 2011 Natural Environment White Paper (which included a Payment for Ecosystem Service Action Plan [31]). It also includes all known privately funded schemes targeting peatland restoration in Europe, where innovative funding mechanisms have proliferated in recent years, providing insights into the operation of ecosystem markets internationally for this important habitat.

## 2 Background

There is a well-known and significant gap between the public funding currently available and the funds that are needed to address the twin challenges of climate change and biodiversity decline [32]. In the UK alone, it has been estimated that it will cost £1.8M to meet Achai biodiversity targets [33], and the cost of reaching net zero GHG emissions by 2050 has been estimated at between £50–70 billion [33]. However, there are significant challenges in delivering emission reductions in the land use sector, where it is estimated that it may cost £247 million to deliver net zero targets from peatlands, woodland and agriculture [34, 35]. This gap is likely to increase as Governments around the world respond to the economic impacts of the COVID pandemic of 2019–20. In the UK land use sector, this is compounded by post-Brexit

agricultural policies, which will lead to an overall reduction in public funding for the sector by 2027 as support moves away from direct payments. The upfront costs of many nature-based solutions are prohibitive for owners and managers in the land use and marine sector, and it can be many years before monetizable benefits accrue, further exacerbating the funding gap.

At the same time, members of the UK Investment Association managed £8.5 trillion in 2020 [36] and the global bond market was worth $21 trillion in 2019 [37]. Within this community is a small but growing group of impact investors who are willing to accept lower than market-rate returns on investment and higher levels of risk [38]. There is also growing recognition in the corporate sector of increasing risks to business from the environment, with climate risks now commonly featuring on company risk registers. Although only 13 percent of US company directors ranked climate change as one of their top five risks for 2020 [39], risk assessments over longer time horizons identify multiple risks from climate change, notably risks from extreme weather to infrastructure and supply chains, and "transition risk" as regulation and consumer preferences shift towards a low carbon economy, amplifying other more traditional risks e.g. being left behind by low carbon technology accelerations and resource scarcity [40].

As a result, demand from the corporate sector is now growing rapidly for ecosystem markets, and there has been a recent proliferation of new schemes and markets, a number of which we review in this paper. These markets are being stimulated by Government investment, with the goal of using public funding to leverage private investment in natural capital. For example, in the UK, a Natural Environment Investment Readiness Fund was launched in 2021 to support the development of projects that can generate revenue from ecosystem services and attract repayable investment. The three-year £10 million programme is providing grants which project developers can use to build capacity and consortia to develop projects to an investible level [41]. The UK will issue its first green government bond in 2021, setting an example to other governments on issuing green bonds in the year that the UK hosts the 26[th] Conference of the Parties to the UN Framework Convention on Climate Change. The UK follows the example of Poland's sovereign green bond (in 2016) and Germany's inaugural green Bund (in 2020).

## 3 Methods

We conducted a comparative analysis of: 1) all known private ecosystem markets operating (or near to market) across dairy, arable, forestry and peatland systems in the UK (Table 1); and 2) all four private peatland ecosystem markets known to be operating in Europe (Table 1). For the purposes of our sample, we defined ecosystem markets as full developed platforms that could facilitate ongoing exchanges between multiple private buyers and sellers of ecosystem services in the UK and in European peatlands. As long as the scheme was designed primarily to facilitate private investment (and this was required in additionality criteria), schemes that also leveraged public investment were included in the sample. Schemes could facilitate investment directly through the purchase of ecosystem services or indirectly by providing credit supply and risk management, as long as the goal of the financial mechanism was to facilitate investment in ecosystem services. Schemes that were deemed out of scope included non-UK schemes (including international voluntary carbon markets), publicly funded schemes that did not require private finance as part of their operation, schemes at concept or early development stages, and single transaction bilateral arrangements that were not part of a longer-term scheme sourcing multiple projects for multiple buyers or investors. For this reason, voluntary and compliance carbon markets were not included in the analysis. Research was conducted in four phases, as shown in Fig 1. Ethical approval was sought and granted from Newcastle University Ethics Committee in May 2018. Informed consent was gained from all participants,

**Table 1. Comparison of UK ecosystem markets and European peatland restoration markets (for additional results and discussion of each row, see S1 File).**

| | UK ecosystem markets | | | | | | | European peatland markets | | | |
| --- | --- | --- | --- | --- | --- | --- | --- | --- | --- | --- | --- |
| | Landscape Enterprise Networks (LENs) | Natural Infrastructure Scheme (NIS) | Woodland Carbon Code (WCC) | Blue Impact Fund | Habitat Banking (HB) | Nature-Climate Bond (NCB) | Natural Capital Pioneer Fund (NCPF) | Peatland Code (PC, UK) | Moor Futures (MF, Germany) | max.moor (MM, Switzerland) | Dutch Green Deal (GDNL, The Netherlands) |
| Status | Operational | Near to market | Operational | Operational | Operational | Near to market | Near to market | Operational | Operational | Operational | Operational |
| Description | Natural capital risks and dependencies are mapped for businesses across a region to create demand-side consortia who invest in soil, water and biodiversity related interventions delivered by landowners across a landscape to reduce risks (e.g. from climate change) and increase resilience of ecosystem services that underpin business operations. Launched initially by 3Keel with Nestle in 2016, the approach is now open source and used more widely. There are currently 8 projects in operation with a further 4 projects under development. Contracts with individual farmers last from 1–2 years, but measures last much longer (such as hedgerow planting) and the LENs scheme itself is open-ended. | A theoretical avoided cost-based model to harness commercial investment in the delivery of natural flood management. Brings together demand and supply side actors to procure natural flood management interventions, that are monitored and maintained for up to 15 years. The scheme was proposed by Green Alliance and the National Trust, but has not yet formally launched or developed any projects. | Voluntary standard for UK woodland creation projects that seeking to build and sequester carbon. The code encourages a consistent approach to woodland carbon projects, and provides assurances to purchases that carbon sold is real, measurable and additional. Launched in 2011 by the Forestry Commission (and now owned by Forest Scotland), there are currently 241 projects validated and a further 154 projects registered but not yet fully validated. Projects last up to 100 years (with the minimum determined by rotation length). | The Blue Impact Fund is a UK-focussed fund seeking to build and enhance the sustainable aquaculture sector through tailored investment and targeting strategic portfolio synergies. The Blue Impact Fund seeks to scale financial, environmental and social outcomes through its investments. The Blue Impact Fund aims to: 1) Protect and restore marine ecosystems e.g. creating no take zones, improving water quality, reducing disease and invasive species, reducing waste and pollution; 2) reduce the climate and ecological footprint of the blue economy e.g. carbon sequestration/ storage, habitat creation, renewable energy; and 3) Improve livelihoods, health and wellbeing for communities e.g. employment/skills, economic growth, locally-sourced and healthy food. The Blue Impact Fund was co-developed by Finance Earth and WWF UK, and launched fundraising in November 2020. The fund has a pipeline of ~ £90 million. The Blue Impact Fund has been created alongside an aligned charity, the Ocean Recovery Trust, that will work to restore ocean health by funding innovation, capacity building, and marine conservation programmes, growing the sustainable blue economy and delivering ocean restoration. The Ocean Recovery Trust will be funded through a 'conservation dividend' generated by the Blue Impact Fund, alongside additional philanthropic donations. | The creation of new woodlands, wildflower meadows, wood meadows, rewilded sites and wetlands is funded by housing, commercial and retail developers that are required to compensate for biodiversity losses via the planning system in each UK country, and to ensure a biodiversity net gain in England and Wales. Launched by Environment Bank in 2006, 19 projects have been delivered to date as bespoke offsets. Larger scale habitat banks are to be rolled out in 2021 to satisfy increasing demand under a mandated regime in planning law. | Abundance Investment and City of Edinburgh Council are developing bonds that could be issued by Local Authorities direct to the public who could invest from as little as £5. This would be facilitated through Abundance Investment's crowdfunding platform to fund interventions such as the creation of urban green spaces or sustainable urban drainage systems. These would provide long-term savings or revenue that would enable Local Authorities to repay investors. | A scheme led by Conservation Capital that is planning to fund biodiversity projects via unsecured loans from impact investors to businesses that are unable to access traditional lending due to the size of their asset base. Profits from these businesses would then enable investors to be repaid with interest. Examples of investments could include nature tourism, oyster reef restoration and development of peat free composts. | Voluntary standard for UK restoration projects that reduce GHG emissions from degraded peatlands. The code sets out best practice requirements and a standard method for the quantification of GHG benefits and provides buyers assurances that the climate benefits being sold are real, quantifiable, additional and permanent. Launched in 2015 by IUCN UK Peatland Programme, there are currently 4 validated projects and a further 6 registered and others in the process of registering. Projects last 30–100 years. | Voluntary standard for German peatland restoration projects based closely on Verified Carbon Standard methodologies. The founders developed the Greenhouse gas Emission Site Type (GEST) approach to assess emissions based on vegetation composition (Couwenberg et al., 2011). Credits cannot be traded on voluntary markets as of now, as they are issued ex-ante. Credits usually sold to private people or German companies wishing to offset emissions. Launched in 2011 in one federal state of Germany, it has now (2020) been taken up by four federal states. There are currently five projects that are validated (three of them verified). Projects last 30–50 years. | A methodology for voluntary carbon market projects restoring Swiss raised bogs. Driven by local authorities and credit developers, and selling to buyers in Switzerland. Targeting lands already retired from agricultural use because of conservation laws in place since the mid 1980ies, but where the rewetting and restoration of vegetation cannot find funding. Launched in 2017 by the Swiss Federal Institute for Forest, Snow and Landscape Research. Projects lasts up to 50 years. | A mutual agreement under private law between a coalition of companies, civil society organizations and local and regional government. The deal defines interventions and outcomes to be delivered and financial and other inputs to be provided by all participants. A three year Green Deal Pilot National Carbon Market launched in 2017, covering forestry, land use change and renewable energy. Projects last 10–50 years. |

(Continued)

**Table 1.** (Continued)

| | UK ecosystem markets | | | | | | | European peatland markets | | | |
|---|---|---|---|---|---|---|---|---|---|---|---|
| | Landscape Enterprise Networks (LENs) | Natural Infrastructure Scheme (NIS) | Woodland Carbon Code (WCC) | Blue Impact Fund | Habitat Banking (HB) | Nature-Climate Bond (NCB) | Natural Capital Pioneer Fund (NCPF) | Peatland Code (PC, UK) | Moor Futures (MF, Germany) | max.moor (MM, Switzerland) | Dutch Green Deal (GDNL, The Netherlands) |
| **Validation and verification of outcomes** | Current projects are focussing on soil carbon, biodiversity, animal health and water quality. Implementation of interventions is validated but outcomes are not verified yet. In future, verified carbon units will be offered via integration of WCC and PC projects as part of a portfolio of benefits across a landscape. | A range of ecosystem services could be delivered, but the focus of the scheme is primarily on hydrological outcomes. No projects have been validated and there is no guidance for project validation yet. There are no plans yet verify outcomes, issue carbon units or enable trading on voluntary carbon markets, and any units generated would not be eligible for the compliance market. | Verified Woodland Carbon Units from WCC projects can be used by companies to compensate for their UK-based greenhouse gas emissions, but cannot be traded on voluntary or compliance carbon markets. A registry enables units to be bought and sold by companies within the UK. Forward selling of Pending Issuance of Verified Carbon Units ex-post. | The Blue Impact Fund was co-developed with WWF UK, who will have two seats on the fund's investment committee responsible for assessing the impact of each investment. The basis for the Blue Impact Fund's impact assessment is the Sustainable Blue Economy Finance Principles, which were developed by the WWF to inform private investment into the blue economy. The Blue Impact Fund's investments will comply with relevant ASC/MSC standards and Sustainable Development Goals (in particular SDG 14 – life below water). These standards will be used to set deal specific KPIs for impact monitoring. | Environment Bank validate their own projects but are designing a third-party accreditation system for on-site and off-site biodiversity net gain delivery. Environment Bank also validate biodiversity net gain calculations made by developers to ensure effective biodiversity accounting. | Not yet established. | Not yet established. | Verified Peatland Carbon Units from PC projects can be reported by companies, but cannot be used as carbon offsets or traded on voluntary or compliance carbon markets. A registry enables units to be bought and sold by companies within the UK. Forward selling of Pending Issuance Units is possible after validation, in addition to the purchase of Verified Carbon Units ex-post. | Carbon Units from MF projects can be reported by companies, but cannot be used as carbon offsets or traded on voluntary or compliance carbon markets. A federal state ministry registry enables units to be bought and sold by private individuals and companies anywhere in the world. Forward selling of Pending Issuance Units is possible after validation, in addition to the purchase of Verified Carbon Units ex-post. | A certificate is issued, which represents remaining peat layer thickness as a proxy for avoided future emissions. Project developers then issue carbon credits based on these, following third part verification following carbon markets standards. | Validation is conducted by a committee of experts, with certification bodies verifying projects based on data collected from monitoring wells. |
| **Additionality and leakage** | No formal additionality tests. The likelihood of leakage is low due to the landscape scale of projects. | No formal additionality tests or consideration of leakage yet. | Additionality tests cover legal compliance, contribution of carbon finance (at least 15%), evidence projects would not have been economically viable without carbon finance, and where the project has overcome other barriers would have otherwise prevented woodland planting (the "barrier test"). Projects have to state any intention to change or intensify land use elsewhere on their holding as a consequence of woodland creation and assess associated GHG emissions. | The Blue Impact Fund has been structured to fill a gap in the market, providing tailored finance and support to businesses that otherwise wouldn't be able to access the same level of support. The BIF will also work with businesses to create portfolio level synergies and work with our pipeline to maximise impact. | Conservation or biodiversity credits generated from bespoke offsets or habitat banks are calculated using the defined Defra metric [47]. The number of biodiversity units generated from a habitat bank take into account the existing condition of the receptor site–of-the receptor site–additionality is therefore guaranteed—credits cannot be generated from land already delivering specific biodiversity value. Nor can high quality habitats be converted into other habitats in order to generate credits. Therefore, leakage is not an issue with habitat banks. | Not yet established. | Not yet established. | Additionality tests cover legal compliance, financial feasibility (at least 15%), evidence projects would not have been economically viable without carbon finance, and where the project has overcome other barriers would have otherwise prevented peatland restoration (the "barrier test"). Projects have to state any intention to change or intensify land use elsewhere on their holding as a consequence of woodland creation and assess associated GHG emissions. | Tests focus on financial additionality (projects must be 100% carbon financed). MF assesses projects for activity shifting, market leakage and ecological leakage, requiring projects with significant leakage to account for these emissions in their assessment of GHG benefits. | Includes a financial additionality test requiring at least 10% carbon finance, and focuses only on degraded peatlands no longer in agricultural use to avoid leakage. | Legal additional test only. Leakage is not assessed in GDNL projects. |

(*Continued*)

Table 1. (Continued)

| | UK ecosystem markets | | | | | | | European peatland markets | | | |
| | Landscape Enterprise Networks (LENs) | Natural Infrastructure Scheme (NIS) | Woodland Carbon Code (WCC) | Blue Impact Fund | Habitat Banking (HB) | Nature-Climate Bond (NCB) | Natural Capital Pioneer Fund (NCPF) | Peatland Code (PC, UK) | Moor Futures (MF, Germany) | max.moor (MM, Switzerland) | Dutch Green Deal (GDNL, The Netherlands) |
|---|---|---|---|---|---|---|---|---|---|---|---|
| **Permanence** | Permanence provided via contractual agreements requiring repayment to buyers if interventions are reversed within the contracted period. Contract lengths are typically short and many of the interventions are easily reversible after the end of a contract, but buyers are aware of this, and primarily procure short-term benefits from the scheme. | No formal mechanisms yet proposed to ensure permanence. | Permanence is provided via contractual agreements and a 15% risk buffer pooled across all projects. Additional protection is provided under the Environmental Impact Assessment (for deforestation) and the Forestry Act (1967) which require a felling licence from the relevant forestry authority to remove trees. | The Blue Impact Fund will offer follow-on investment during the fund's life to further grow and optimise its investments. The investments targeted are operational businesses so will continue deliver impact beyond the funds exit. The fund's exit strategy will be determined after the optimisation phase of the portfolio, which seeks to maximise the operational and portfolio efficiencies, but is likely to comprise either a public listing, which would enable the fund to structure long-term impact objectives within the portfolio structure, or strategic market sale. In each case, the fund will consider permanence within the decision-making. | Provided by mandated 30-year contracts between purchaser/broker or other authority and landowner provider. Environment Bank Conservation Bank uses 30-year Conservation Bank Agreements (legal agreements between Environment Bank and the landowner provider) with annual payments to landowners according to delivery of key milestones in the Agreement in association with an agreed payment plan, as verified via regular monitoring and reporting. | Not yet established. | Not applicable. | Permanence is provided via contractual agreements, which require projects to compensate buyers for reversals. Further guarantees can be provided via Conservation Covenants in England and Conservation Burdens in Scotland, which place a requirement to maintain projects on anyone who purchases the land. Permanence is also ensured via a 15% risk buffer and 10% precision buffer of unsold carbon units pooled between all projects. | The permanence of MF projects is provided through contracts, prescribed water levels under the Water Law, entries in the land register to secure permanence of the required water levels and/or the purchase of land for restoration through a Trust that can guarantee the long-term maintenance and management of project sites. Permanence is also ensured via a 30% project buffer of unsold carbon units. | Permanence is included in contracts and increased by only focusing on degraded peatlands that are no longer in agricultural use, but there are no formal measures to guarantee this. | Permanence is not assessed in GDNL projects. |
| **Supply and demand issues** | Prices determined through negotiation between buyers and sellers via supply aggregators, based on costs and ecosystem service revenues. Payments made annually via a regional legal entity under development. Prices are based on a bundle of ecosystem services typically including soil function, water quality and biodiversity, but these outcomes are not typically quantified. Demand is typically constrained to businesses with an interest in the LENs landscape | Prices determined through negotiation between buyers and sellers via supply aggregators, based on costs and ecosystem service revenues. Prices and payment mechanisms have yet to be reached. | Prices determined through negotiation between buyers and sellers, either directly or via intermediaries, based on costs and ecosystem service revenues. Upfront payments for woodland establishment followed by annual payments, primarily on the basis of carbon sequestration, though a Wider Benefits Tool enables estimation of co-benefits. Availability of land for afforestation is increasingly likely to constrict supply of projects to the Code. | The Blue Impact Fund doesn't foresee any material issues in the supply of projects: the market is fast-growing and the most impactful concept/scale up phase; particularly within the seaweed sector. COVID has had an impact on the seafood sector and presents an opportunity to bring investment into distressed businesses that demonstrate sustainable growth potential (e.g. through consolidation/expansion). | Prices of conservation credits currently set for habitat banks by Environment Bank for establishment and 30-year management. Planning Authorities are key to ensuring demand for developments to deliver at least 10% Biodiversity Net Gain. A mandated system will give clarity and certainty, reduce planning delays and increase demand, and it is not yet clear if there will be sufficient supply to meet this new demand. Planning Authorities may use their own land but legally challengeable. | Not yet known. | Not yet known. | Prices determined through negotiation between buyers and sellers, either directly or via intermediaries, based on costs and ecosystem service revenues. Upfront payments for restoration capital works, followed by annual payments, primarily on the basis of avoiding GHG emissions, though co-benefits may also be marketed. There are currently a number of issues constraining the creation of projects to supply current demand. | Prices are fixed on the basis of project costs. Upfront payments for restoration capital works, followed by annual payments, primarily on the basis of avoiding GHG emissions, though detailed guidance is provided to assess co-benefits, which may also be marketed. | Prices are fixed on the basis of remaining project costs after public investment. Upfront payments for restoration capital works are paid from public funds with private investment supporting annual payments, on the basis of avoiding GHG emissions. | Prices determined through negotiation between buyer and seller. Payments for units expectedly ex-post, but not regulated. In Q2 2020, a handful of projects were under preparation, using the peatlands methodology but no units issued yet. |

(Continued)

Table 1. (Continued)

| | | | UK ecosystem markets | | | | | | European peatland markets | | |
|---|---|---|---|---|---|---|---|---|---|---|---|
| | Landscape Enterprise Networks (LENs) | Natural Infrastructure Scheme (NIS) | Woodland Carbon Code (WCC) | Blue Impact Fund | Habitat Banking (HB) | Nature-Climate Bond (NCB) | Natural Capital Pioneer Fund (NCPF) | Peatland Code (PC, UK) | Moor Futures (MF, Germany) | max.moor (MM, Switzerland) | Dutch Green Deal (GDNL, The Netherlands) |
| Interaction with public funding | Projects are covered entirely by private investment with very limited, indirect Government support for scheme development and operation. | Projects envisaged to be covered by private investment with only limited, indirect Government support for scheme development to date. | Up to 85% of project costs can be publicly funded. Development and operation of scheme is primarily publicly funded. | The Blue Impact Fund is a private equity model but is open to interacting with public sources through match funding, grant and other resource support. The Ocean Recovery Trust, aligned to the Blue Impact Fund, will target engagement with the public sector to leverage additional investment for the ocean economy. | At present Biodiversity Net Gain, whilst already scaling, has failed to attract Government funding. | Local government would be the issuer of the bonds and is therefore a key player as the entity that would take on the risk. Interventions would be funded by private investment. | Impact investors would provide the majority of capital, supplemented by grant funding in the early stages. | Up to 85% of project costs can be publicly funded. Development and operation of scheme is primarily publicly funded. | Projects are covered entirely by private investment, with significant, indirect Government support for scheme development and operation. | Up to 90% of project costs can be publicly funded. Development and operation of scheme is primarily publicly funded. | Up to 85% of project costs can be publicly funded. Development and operation of scheme is primarily publicly funded. |

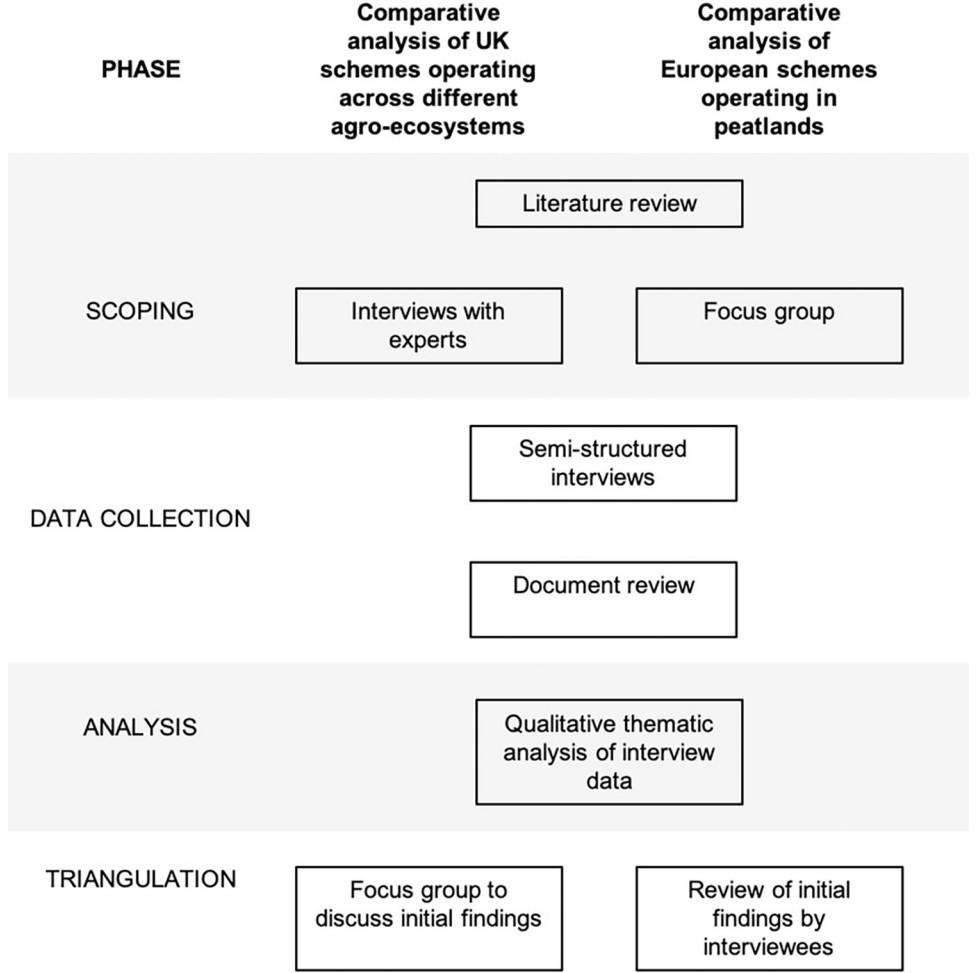

**Fig 1.** Research design showing different phases of the research showing where identical methods were used (centre) or where different methods were used in each phase for the UK comparison of PES schemes across different agro-ecosystems (left) and the comparison of peatland schemes across Europe (right).

documented via signed consent forms with accompanying participant information sheets. For more detailed methods, see Gosal et al. [42] and Olesen et al. [43].

## 3.1 Phase 1: Scoping

A narrative literature review was conducted to identify all private ecosystem markets currently operating or near to market in the UK (in any agro-ecosystem) and all private ecosystem markets operating in peatlands in Europe. Unlike systematic reviews or meta-analyses, a narrative literature review is an expert-based "best-evidence synthesis" of key literature [44], which is better suited to reviews that aim to provide a broad overview via expert synthesis, where it is difficult to identify specific outcome measures [45]. This literature also served to identify interview topics and questions for Phase 2. To ensure all relevant schemes were identified (including those close to market, which were not in the public domain) and refine the parameters of the analysis, scoping interviews were conducted for the UK comparative analysis and the two case studies, and a focus group was conducted with seven participants (including researchers, consultants and EU policy stakeholders) for the European comparative analysis of peatland

schemes. During this phase, a number of new schemes were identified for analysis or removed from the study, on the basis of the inclusion and exclusion criteria in the previous paragraph.

### 3.2 Phase 2: Data collection

Data was collected in 2020 via a review of documentation for each scheme and semi-structured interviews with scheme representatives and intermediaries, which covered governance and legal matters, economics and funding and the operation of each scheme. For the UK, 12 interviews were conducted with representatives the eleven markets: the Woodland Carbon Code (WCC), Landscape Enterprise Networks (LENs), Habitat Banking (HB), the proposed Natural Infrastructure Scheme (NIS), Nature-Climate Bond (NCB), Natural Capital Pioneer Fund (NCPF), Habitat Banking (HB) and the Blue Impact Fund (BIF). For the European peatland market analysis, a further 13 interviews were conducted with representatives of four private peatland ecosystem markets (the PCC in the UK, MoorFutures (MF) in Germany, max.moor (MM) in Switzerland and the Dutch Green Deal (GDNL).

### 3.3 Phase 3: Analysis

Qualitative data from interview and focus group discussions were analysed thematically alongside documentation from each scheme. Interviews were recorded, transcribed and anonymised in line with ethical approval from the Newcastle University. The thematic analysis approach outlined by Braun and Clarke [46] was used to undertake in-depth analysis of the interview and focus group transcripts in three stages: initial coding of ideas, views and concepts into minor themes; review and refinement of minor themes to identify major themes; evaluation of themes in relation to the objectives of the study to draw in relevant insights to the comparative analysis [47]. Theoretical saturation was considered to be achieved when no new themes were identified from transcripts.

### 3.4 Phase 4: Triangulation

Finally, preliminary findings from interviews and review of scheme documentation were triangulated via individual written feedback from interviewees (with those providing extensive inputs offered co-authorship), with the addition of a focus group for the UK schemes. In the focus group, findings from the interview phase were presented to participants for discussion in plenary, before breaking into two parallel groups to discuss options for integration between the three main private schemes in operation in the UK, and between public and private schemes. The focus group was attended by 12 participants including researchers, consultants, businesses, the third sector, an intermediary/broker and policy stakeholders from regulatory bodies and Government departments in Scotland and England.

## 4 Results

Table 1 describes and then compares each of the schemes reviewed in terms of their approach to: validation and verification of outcomes; additionality and leakage; permanence; supply and demand issues; interaction with public funding; and scheme governance. These are discussed further in the Supplementary Material, Gosal et al. [42] and Olesen et al. [43].

The comparative analysis identified a number of points of commonality between the schemes that were reviewed (see S1 File and Table 1 for more details). In summary, common characteristics and challenges across all schemes are:

- Participation in all schemes was voluntary for both buyers and sellers. Clearing prices were reached between buyers and sellers occurred in a minority of schemes, mainly regional

ecosystem markets. For the majority, prices were primarily determined by project costs, which were highly variable both within and between schemes. None based their prices on the price per tonne on the voluntary carbon market, which would typically have been too low to cover project costs. One of the ways that the four peatland schemes justified higher prices (compared to international carbon market prices) was by highlighting additional non-carbon benefits such as water quality benefits of restoration or biodiversity (more on bundling versus stacking of multiple services below). The marketing of co-benefits was common across all the schemes reviewed, but verification and quantification of co-benefits were limited in nearly all schemes (MF being the exception).

- Most schemes used intermediaries to engage with project developers (e.g. landowners and tenants), or the scheme itself performed this function (e.g. BIF) and LENs used supply aggregators to aggregate sufficient density of supply within a single landscape. However, engagement with suppliers (typically landowners and managers) was a challenge for many schemes. In contrast, BIF had created a £90M project portfolio prior to entering its investment phase and did not foresee issues meeting demand from investors.

- On the demand side, sensitivities around the willingness of businesses to share financial data were identified as a challenge to the establishment of co-procurement arrangements between buyers in schemes where this was possible. As well as this, additionality was an issue for buyers in some schemes where businesses were reluctant to pay for interventions that landowners/tenants should already be doing to comply with regulation and/or that could be paid for by public finance.

- Across the schemes, consideration of the wider social distribution of ecosystem services was limited, although there was recognition of its importance for buyers with Corporate Social Responsibility goals.

- Permanence was addressed primarily via contractual arrangements in the schemes reviewed, although Conservation Burdens (Scotland) and Covenants (England and Wales) were sometimes proposed by schemes as potential future options to provide additional assurances to buyers in some UK schemes, and BIF provided follow-on funding opportunities to enhance permanence.

In addition to these common characteristics and challenges, the comparative analysis identified a number of important differences between the schemes that were reviewed (see S1 File and Table 1), for example:

- The four peatland schemes and WCC tended to validate and verify outcomes using site visits by independent certification bodies, HB was developing a third-party accreditation system and BIF accredited projects to relevant industry standards. However, validation mechanisms had not yet been developed for NCB and NCPF, and LENs and NIS provided validation in the form of evidence that interventions had been carried out, without requiring independent verification of ecosystem service outcomes.

- Additionality was only assessed formally by the four peatland schemes, WCC and HB, typically via legal (e.g. projects go beyond what would be required by law), financial (e.g. projects would not be possible without carbon finance) and other additionality tests (e.g. application of biodiversity metrics in HB receptor sites). None of the other schemes applied formal additionality tests, relying instead on trusted intermediaries to manage additionality informally as part of the project design process (e.g. LENs) or identifying businesses that had been unable to fund sustainability initiatives via other means (e.g. BIF).

- The peatland schemes and WCC tended to focus on selling (often fungible) climate mitigation benefits via market registries (e.g. the UK Land Carbon Registry run by IHS Markit). While co-benefits, such as biodiversity benefits were used to market these schemes as part of a bundle of services anchored on carbon, only MF quantified these benefits as part of an explicit package of multiple ecosystem services that were all being sold together. In contrast, other schemes were designed to sell or finance a wider range of (mainly non-fungible) ecosystem services, including water quality, soil function, biodiversity and animal welfare benefits, in addition to climate mitigation to buyers. None of the schemes stacked different fungible services for sale to different buyers. Additionality rules of fungible schemes meant that stacking of fungible services provided by the schemes reviewed would only be possible where the cost of delivering the service was too high to be financially feasible through the sale of a single service (e.g. the price per tonne of carbon would be prohibitive). However, where interventions deliver more than one service, and neither service could bring in sufficient funding to cover the cost of the intervention, stacking would in theory meet additionality tests in each scheme. For example, stacking could enable projects to meet "financial feasibility" tests where multiple sources of ecosystem service payments were necessary to fund the minimum threshold for private finance (15% in the case of PC and WCC). Alternatively, "economic alternative" tests could be met where the project would otherwise not be the most economically attractive option for that location, for example carbon finance alone is unlikely to outweigh the opportunity costs of replacing arable agriculture or horticulture with paludiculture or habitat restoration in lowland peat fenlands, but the addition of biodiversity finance might make habitat restoration economically attractive as an alternative to current land use. Finally, "barrier" tests could be met if it can be shown that the project would not be possible for any other reason without stacking payments for multiple ecosystem services.

- Investments in the peatland schemes, WCC and HB tended not to be geographically linked to the locations or interests of buyers, who they sourced nationally, and some of these schemes ruled out international investment to avoid double-counting against national emission reduction targets. LENs, NIS, BIF, NCB and NCPF were able to accept funding from national and international buyers (e.g., overseas impact investors). However, LENs and NIS tended to focus on sourcing funding from regional stakeholders, on the basis that this is a scale at which synergies and benefit integration are easier to achieve.

- Schemes relied to varying extents on public funding, both in terms of scheme operation and project financing. The peatland schemes and WCC were significantly more reliant on public funding for project financing and in many cases scheme operation than the other schemes reviewed.

- Payment mechanisms varied significantly across schemes (and in some cases between interventions within schemes) with the use of different legal agreements, payment structures and investment aggregation platforms (ranging from intermediaries acting as demand aggregators to crowdfunding platforms).

## 5 Discussion

In this section, we will discuss some of the key differences between the schemes and markets included in the comparative analysis and explore the potential to integrate different types of ecosystem markets. In doing so, we explore the governance issues associated with private market integration and the blending of private and public funding for public goods.

**Table 2. Typology showing the defining characteristics of national carbon markets, regional ecosystem markets and green finance.**

| Defining characteristic | National carbon market | Regional ecosystem market | Green finance |
|---|---|---|---|
| Main benefits for investors | Climate mitigation benefits, sometimes offsets | Management of environmental risks to the delivery of business objectives | Economically sustainable delivery of public goods from private finance that can deliver returns on investment |
| Verification and validation of projects and outcomes | Strict procedures governing validation of projects and verification of outcomes by independent certification bodies | More limited verification of outcomes, including by projects themselves and intermediaries | Verification by scheme operators or independent bodies to industry or Government agreed metrics or standards |
| Additionality | Assessed formally at project scale via legal, financial and other additionality tests with limited consideration of landscape scale effects sometimes via leakage assessments | Assessed informally at landscape scale by intermediaries during scheme design to avoid ecosystem service trade-offs and free-riding | Assessed informally during the construction of the project pipeline or formally via metric-based additionality tests on site |
| Ecosystem service outcomes | Focus on selling (often fungible) climate mitigation benefits via market registries | Sold a wider range of non-fungible ecosystem services, including water quality and quantity, soil function, biodiversity and animal welfare benefits, in addition to climate mitigation, which were often bundled together in integrated landscape scale propositions | Financed the widest range of ecosystem services, including prevention and removal of invasive species, urban green space, sustainable urban drainage systems and development of peat free composts, some of which were fungible. |
| Operating scale and market scope | Landscape scale projects typically offered nationally to buyers anywhere in the country, often not allowing international buyers to participate to prevent double counting against national emission reduction targets | Landscape or regional scape projects typically developed for buyers within the same region, although national and international buyers can in theory participate | Landscape or regional business scale projects developed for national and international impact investors and members of the public |
| Reliance on public funding for projects and/or scheme operation | Significant (up to 85% project costs) | Limited | Limited |
| Examples (for details, see Table 2) | Woodland Carbon Code Peatland Code MoorFutures max.moor Dutch Green Deal | Landscape Enterprise Networks Natural Infrastructure Scheme | Nature-Climate Bond Natural Capital Pioneer Fund Habitat Banking Blue Impact Fund |

## 5.1 Types of ecosystem market

Based on the comparative analysis in Table 1 and Supplementary Material, three different types of scheme emerged, based on their modes and geographical scales of operation, funding and outcomes (Table 2):

1. **National carbon markets**, primarily sold verified, validated, additional and (often) fungible climate mitigation benefits (sometimes marketed as offsets), typically to national buyers within a single country at prices that reflect project costs more than they reflect carbon market rates, with permanence provided by legislation or long-term contracts and significant Government funding for projects and/or scheme operation. These differ from international voluntary carbon markets, which allow international buyers to purchase and trade carbon at market rates, and from compliance markets which are regulated by mandatory national, regional, or international regimes and only allow trading between regulated entities;

2. **Regional ecosystem markets** enabled buyers to manage environmental risks to their business by investing in a wider range of non-fungible ecosystem service outcomes (including water quality, soil function, biodiversity and animal welfare benefits), in addition to climate mitigation, typically to regional buyers, with varying levels of validation, verification, additionality and permanence and limited Government funding required for projects and/or scheme operation; and

3. **Green finance** provided risk-adjusted returns on investment for national and international investors (potentially including members of the public via crowdfunding) who were willing

to accept lower than market rate returns, and financed the widest range of (sometimes fungible) ecosystem service outcomes, with verification of outcomes (and in one case additionality) using industry or Government agreed metrics and standards, permanence via long-term contracts or follow-on funding and limited Government funding required for projects and/or scheme operation.

Although not included in our sample of UK-based schemes that are operational (or close to market) and peatland schemes in Europe, green finance mechanisms can also include loan-based schemes and insurance products. For example, Scottish Natural Heritage, Scottish Environmental Protection Agency, Scottish Wildlife Trust, RSPB, British Ecological Society and British Marine are developing a scheme based on loans with Lloyds Bank, in which commercial bank loans are be made to groups that can implement biosecurity measures to prevent the arrival or spread of invasive species or help eradicate them. Loans would be repaid from future savings on the costs of managing invasive species [48]. For example, Willis Towers Watson have a Global Ecosystem Resilience Facility uses the prospect of reduced premiums to encourage investment in projects that reduce climate-related and other environmental risks to clients (e.g., coral reef protection and restoration to protect coastal businesses from storm surges), reducing risks and so making premiums more affordable [49]. Corporate Social Responsibility (CSR) schemes are not included in the typology, as this is one of a range of motives for investing in ecosystem markets, and CSR can motivate investment in all three types of scheme identified above.

The typology in Table 2 provides an evidence-based distinction between the key types of ecosystem markets operating in the UK and Europe on the basis of their modes of operation, funding and outcomes. This may provide useful clarity for decision-makers in policy and practice who wish to expand the role of private investment (ranging from crowdfunding to green investment funds) in conservation and regenerative agriculture. For example, a practitioner may be able to use the typology to identify relevant existing schemes or develop new schemes that target the types of ecosystem services, project developers or investors they are most interested in. Alternatively, a policy-maker targeting climate change mitigation from the land use sector might prioritise the promotion of national carbon markets, whereas a Local Authority seeking to reduce flood risk might prioritise paying for or attracting private investment in natural flood management via LENs and/or investment in sustainable urban drainage systems via green bonds or other green finance mechanisms. A decision-maker interested in providing additional income streams for farmers might focus on a peatland scheme or LENs, and if they wanted to exclude overseas investment to ensure investments counted towards national net zero targets, they might focus on national carbon markets and regional ecosystem markets, rather than green bonds which tend to attract international impact investors. The typology also provides new academic insights into the operation of ecosystem markets in practice, which may challenge traditionally held notions of PES. In particular, regional ecosystem markets do not conform to a number of assumptions underpinning PES and financial markets, in which payments would normally be conditional on, or linked to, ecosystem service outcomes or returns on investment. For this reason, the next section considers the operation of this type of ecosystem market in greater depth.

## 5.2 Understanding the success of regional ecosystem markets

The emergence and successful early operation of the regional ecosystem market model is particularly noteworthy, given how differently this model operates compared to national carbon markets and green finance (Table 2). What constitutes a PES and how to define it is subject to

much debate [50], but generally there is agreement on PES involving individuals or organisations ('buyers') paying other individuals or organisations who manage natural resources ('sellers') to deliver clearly defined benefits or "ecosystem services" from nature [14]. While this definition of PES relaxes Wunder et al.'s [9, 10] original stipulation that transactions must be voluntary (they rarely are in publicly funded PES schemes), the conditionality of payments on the delivery of well-defined, agreed outcomes remains central to PES, and is widely assumed to be necessary to engender the necessary buyer confidence to underpin a functional ecosystem market. Therefore, the limited provisions for validation, verification and additionality in the regional ecosystem markets reviewed in this research may either be used to question whether these are indeed PES schemes, or to question how important conditionality is to the success of a PES scheme. Moreover, unlike green finance schemes, regional ecosystem markets are not designed to provide returns on investment.

As such, it may at first glance seem surprising that the LENs scheme in particular had attracted significant levels of private sector investment and was proliferating across UK landscapes with new international LENs propositions being developed at the time of the analysis. Although prices across the schemes reviewed were dictated primarily by the costs of delivering projects and so varied from project to project, national carbon markets tended to calculate the cost of projects per tonne of carbon as a reference point to guide buyers. In contrast, LENs buyers had no way of knowing the likely climate benefits, let alone the price of these per tonne of carbon. Instead, they took a more risk-based approach to negotiating prices between buyers and sellers on the basis of risks to assets, supply chains or reputational risks, which could be reduced or avoided by paying for interventions in the landscape. In addition to providing a different reference point for buyers in the negotiation, the focus on risk often brought more senior executives responsible for risk management to the negotiation table with access to different budgets, compared to the sustainability and corporate responsibility officers typically involved in decisions around carbon offsetting. In addition, the metrics typically used to assess risk tended to be less precise than those used to assess offsetting, which may explain the willingness to work with trusted intermediaries to deliver risk reduction outcomes without the controls on verification, validation and additionality of projects that were seen in national carbon markets.

This focus on risk management may also explain the broader range of interests captured by regional ecosystem markets, including for example, asset risks from flooding, supply chain risks arising from issues with water quality, soil function or animal welfare, reputational risks arising from threats to biodiversity, and the wider risks to assets, supply chains and reputation arising from climate change. This diversity of interests then drove demand for multiple ecosystem services, which had to be managed in space and time to avoid trade-offs where the delivery of one service (e.g. carbon via conifer plantation) compromised the delivery of another (e.g. biodiversity). However, this diversity of outcomes also increased the likelihood that companies who did not invest in a scheme may benefit from the investments of their competitors (the free-rider effect). The LENs scheme addressed the challenge of avoiding both trade-offs and free-riders by identifying multiple risks across landscapes used by a number of beneficiary organisations who could manage risks by working together at landscape scales. This increased the number of co-investors to reduce the free-rider effect whilst ensuring interventions worked together without generating trade-offs at the landscape scale through the identification of multiple interests across the investor community prior to constructing the landscape scale interventions to deliver against those interests. Aggregating demand for ecosystem services in this way also increased the overall amount of funding (by stacking payments for multiple benefits) and led to perceptions of long-term resilience in funding, as the risks of any individual investor withdrawing funding were reduced with an increased number and diversity of investors. This

is consistent with the definition of a place-based PES scheme [13], which emphasises the multi-level governance of social, economic and biophysical attributes that shape a given place by bundling or layering the widest relevant range of ecosystem services in the same landscape. The successful aggregation of demand in LENs was in part due to the proactive role of trusted business-to-business brokers, compared to the national carbon markets, which tended to be managed by organisations with very different cultures and language (typically Non-Governmental Organisations, research institutes or Government agencies), who played a more passive role in engaging with investors.

On the supply side, the limited requirements around verification, validation and additionality had the benefit of reducing red tape for land managers who wished to engage with regional ecosystem markets. Indeed, evidence from interviews with farmers working in the LENs scheme in Cumbria have shown widespread satisfaction with the scheme compared to the complexity of public agri-environment schemes [51]. Although farmers still commented on the additional reporting burden, and had other criticisms of the scheme, engagement with the scheme was strong. The two most important drivers for farmer engagement with the scheme, according to a subsequent Delphi survey, were: i) additional, stable income for easily planned and reported, and flexible activities that were compatible with existing management; and ii) improving environmental outcomes and animal health [52]. In addition to the relative simplicity of the regional ecosystem market model, trusted intermediaries were employed to actively recruit farmers, further reducing barriers to entry. These intermediaries aggregated suppliers of services, and so increased market potential (availability of saleable benefits) while reducing transaction costs (of contracting with multiple landowners/tenants).

In contrast, national carbon markets were less proactive in recruiting land managers to develop projects. Neumann [53] conducted a Social Network Analysis of PC and MF, showing little or no engagement with land management representatives in the two governance networks. Instead, decision-making was primarily taken by scheme co-ordinators in consultation with a small number of key researchers who acted as knowledge brokers, providing access to necessary evidence. There was limited active involvement from members of the policy community, although interviews showed that "weak ties" in the network to these more peripheral actors had played an important strategic role in gaining political support and funding for the two schemes. The role of the most engaged researchers in both networks was multifaceted, acting as trusted intermediaries to members of the policy community as well as providing access to evidence to inform scheme development and management. However, both networks were highly dependent on the knowledge, experience and trust that had been accumulated by a small number of scheme co-ordinators and managers, making the ongoing success of both schemes vulnerable to the impact of staff turnover (indeed, the Peatland Code Manager was replaced soon after the research was conducted). In the case of the Peatland Code, the Director had similarly strong networks, providing a degree of resilience to the management of the network. In the case of MF, despite stronger reliance on a single scheme co-ordinator and additional scheme coordinators in the other participating federal states, a larger number of researchers and practitioners played pivotal roles in the network, which may provide this scheme with more resilience to changes in staffing, compared to the Peatland Code. Despite the relatively informal governance arrangement of MF, compared to the two formal governance structures in the Peatland Code, the day-to-day operation of both schemes was similarly dependent on a small number of active members who regularly exchanged knowledge with others, and who were trusted by their network.

More complex and formalised governance structures may be necessary to ensure accountability and transparency as new regional ecosystem markets develop and seek integration with national carbon markets. However, the successful operation of these schemes needs to mitigate

the risk of losing key trusted individuals from the network, if these individuals provide access to expertise, political capital, funding and experience from across their networks. Equally, scheme resilience and delivery of outcomes may be strongly influenced by a small number of key players, which may limit the rate at which new schemes can successfully proliferate, based on their individual capacity.

## 5.3 Integrating private schemes

The main reasons for integrating national and regional ecosystem markets that emerged from the stakeholder workshop (see phase 4 in Methods) were to increase levels of investment and drive more multifunctional outcomes from landscapes. National carbon markets have the potential to bring in new investors to regional ecosystem markets from beyond the region, and regional ecosystem markets have the potential to extend the range of habitats, land uses and interventions that can be funded, beyond those currently covered by national carbon markets. There is a danger that single habitat/service schemes, such as woodland carbon schemes may drive certain outcomes (e.g. climate change mitigation) at the expense of others (e.g. biodiversity), but by integrating national carbon markets and regional ecosystem markets, it may be possible to aggregate demand across multiple habitats and land uses for multiple ecosystem services, and so design schemes that reduce the likelihood of ecosystem service trade-offs.

However, there are a number of governance and technical (e.g., additionality) challenges involved in integrating ecosystem markets. Integrating schemes could generate unwelcome levels of complexity, compared to retaining the status quo of separate schemes, given that these schemes are already operational without integration. There is also a danger that the "commercial force" of carbon markets (as one private sector stakeholder put it) might disrupt regional ecosystem markets that are not currently tapping into this market, leading to a significant refocussing of attention on a single ecosystem service.

The need for schemes to deliver additional outcomes that would not otherwise have been delivered (or legally required) poses a more significant challenge to the integration of national carbon markets and regional ecosystem markets. As described in Section 3.2, regional ecosystem markets were less likely to include formal additionality tests, relying instead on quality assurance of work undertaken to deliver outcomes. However, if income streams for climate mitigation via a national carbon market are integrated with funding for a wider range of ecosystem services via a regional ecosystem market for a package of linked interventions on the same parcel of land, it may be difficult to ensure additionality tests are met. For example, if payments for water quality improvements are stacked on top of carbon payments for a peatland restoration project, it may be difficult to prove that the restoration would not have happened without the carbon finance. One solution to this is for national ecosystem markets to apply financial additionality tests (e.g. the Peatland Code and Woodland Carbon Code require a minimum of 15% carbon finance to be additional). In the case of the Woodland Carbon Code, projects can be de-registered if they integrate additional funding that was not declared prior to validation. Alternatively, although complex and currently untested, it may be possible to apportion credits to different budget contributions within a single project, limiting carbon credits to the proportion of the project funded by carbon finance. The simplest solution however, currently being pursued by UK schemes, would be to spatially separate the delivery of ecosystem services from different schemes, for example integrating peatland restoration and woodland creation in different locations upstream from farm-based projects managing soil carbon or planting hedgerows.

The importance of intermediaries and brokers in achieving integration cannot be understated. In addition to working as supply and demand aggregators (see previous section), they

also play an important role in identifying interventions and projects that could deliver monetizable benefits, demonstrating cash flows, evaluating risks and calculating potential return on investment, before presenting opportunities to investors, where relevant accrediting projects to standards (like those developed for national carbon markets) to increase investor confidence [54]. Evidence from the comparative analysis suggested that communication and trust between scheme actors may be as important as the development of formal governance structures. For two of the peatland schemes analysed (PC and MF), there was evidence that researchers may play a more important role than has been previously appreciated [53], as trusted knowledge brokers and advocates rather than simply as providers of knowledge and evidence (c.f. Pielke [55]). Financial brokers have the capacity to work across all three types of ecosystem market, and initiatives like the Broadway Initiative, Green Finance Institute and SRUC's Thriving Natural Capital Challenge Centre in the UK are already connecting many private schemes and working with Government to create an enabling policy environment.

Building this discussion, Fig 2 proposes three ways in which transactions between buyers and sellers could be managed to integrate both national carbon markets and regional ecosystem markets. In **Option 1**, a regional ecosystem market procures climate mitigation benefits from a scheme that is also supplying national carbon markets or green finance markets, either directly via a demand aggregator or intermediary (A), or with the demand aggregator procuring ecosystem services as part of a package of benefits arranged by a supply aggregator/intermediary (B). In **Option 2**, the carbon or green finance scheme acts as the supply aggregator, providing multiple functions from its own scheme with options (C) to source interventions from other supply side entities. The scheme may also supply additional climate mitigation benefits into national markets (D). In **Option 3**, the carbon or green finance scheme provides both demand and supply aggregation functions. Although this is the simplest integration option, it creates a conflict of interest because the same body is negotiating on behalf of both supply and demand side parties to the transaction. An important principle in integrating carbon, or any additional function into a multifunctional landscape trade, is that different income streams should be put together simultaneously to make a trading proposition, and that the full range of ecosystem services to be provided should be agreed prior to the proposition being agreed and implemented. Once implemented, there is typically little incentive for future buyers to pay for outcomes, since those outcomes are already being delivered, and the additionality tests of national carbon markets would not be met, since activities on the ground would demonstrably not be dependent on the additional payment.

## 5.4 Options for blending public and private finance for ecosystem services

Finally, the research highlighted a number of potential areas of conflict between public funding for natural capital and privately funded ecosystem markets. These included the potential for public funds to outcompete private funds (e.g., where public schemes offer more attractive terms including shorter contract lengths and simpler or more familiar application processes), that would otherwise have enabled the market to deliver the public good. There was also considerable uncertainty over future public schemes as the UK develops and trials post-Brexit policy over a relatively long time-frame, which could freeze the market, with potential sellers withholding projects until they know whether they will get a better price or terms under existing private schemes versus future public schemes.

To tackle these potential conflicts between public and private finance, three broad approaches may be considered:

1. Full public-private co-procurement of public goods, in which public and private finance are integrated into a single fund at a landscape scale designed to deliver multiple outcomes. An

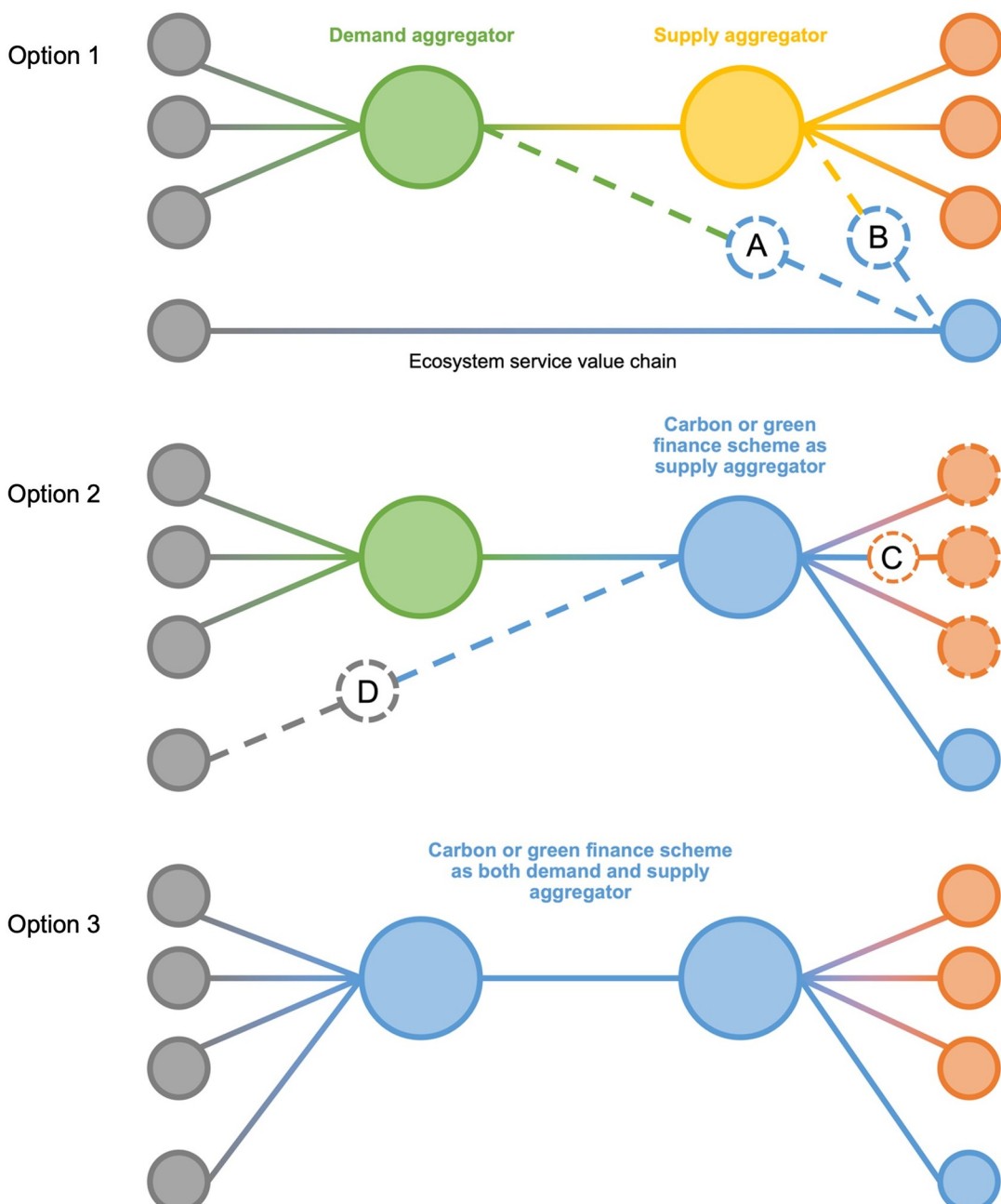

**Fig 2. Three alternative options for integrating national carbon markets and green finance schemes with regional ecosystem markets, showing the different roles each type of market could play in the aggregation of supply and demand for ecosystem services.** Grey = demand side interests, Green = a demand aggregator, or buyer-group, Yellow = supply aggregators, Orange = individual suppliers (often farmers), Blue = a carbon scheme / operator.

example of a private-led integration mechanism would be LENs, which could be adapted to integrate public funding as part of its demand aggregation process, co-ordinating landscape outcomes across multiple private and public investors. An example of a public-led integration mechanism like this would be Rural Land Use Partnerships, which are being piloted by Scottish Government to enable rural communities to shape natural capital investment priorities and provide benefits to these communities alongside land managers and other

providers of ecosystem services. The advantages and disadvantages of this option are outlined in mechanism 1 in Table 3;

2. Co-ordinated public-private funding of public goods, delineated in space or time, enabling the market to pay for as much as possible, while public payments focus on market failures and those who are not prepared to accept private finance. There are a range of mechanisms that could facilitate this, which are outlined in mechanisms 2–6 in Table 3; and

3. Private funding pays for services that are not already being procured by public funding, with limited coordination. This is the "business as usual" scenario in most countries, and the primary mechanism that facilitates this are the additionality criteria in national carbon markets that enable public funding of schemes up to a certain amount (e.g. 85% in the Woodland Carbon Code and Peatland Code).

The six options described in Table 3 and Fig 3 show how public funding might be designed in future to incentivise participation in privately funded PES schemes, enabling the market to deliver significantly more public goods than it currently provides, while reserving public funding to address market failures and avoid distributional justice concerns about inequitable benefits arising from an entirely market-driven system.

Several of these approaches may work best in combination. For example, funds delineation (Table 3) prioritises projects for the market that are able to deliver the most in-demand ecosystem services at the lowest price (often climate mitigation benefits), reserving public funds to pay for projects that are more expensive per tonne of carbon, but that offer other important ecosystem services that have a high value to society e.g. biodiversity or recreational benefits. A cost-benefit matrix (Table 3) or decision support tools such as the tool developed by Artz et al. [56] for Scottish peatlands, could be used to identify sites most likely to deliver cost-effective GHG emission reductions based on the level and type of degradation, and factors likely to influence the cost of restoration, such as accessibility. At the same time, this tool could be used to delineate sites that would be more expensive to restore, but where there may also be important biodiversity and water quality benefits, reserving these sites for investors more interested in these outcomes, and prioritising public funding to sites and/or ecosystem services that the market fails to deliver. Where schemes do not allow overseas investment, the climate mitigation benefits of these privately funded projects count towards domestic net zero targets. However, where overseas investment is permitted, a balance has to be struck between the need to use public funding to meet net zero targets (and so designing public funding schemes to compete with private markets for the most cost-effective sites) versus prioritising sites where market failure is most likely to result in a lack of funding for public goods from nature. Funds delineation is relatively straightforward to implement (compared to many of the other options discussed below), but there is a danger that making room for private markets in this way doesn't leverage additional private finance in some of the ways that other approaches can. Having said this, funds delineation is likely to stimulate some additional private investment by removing the option of public funding for the sites that are most attractive to the market, ensuring public schemes do not compete with private schemes, and increasing the number of projects that are therefore offered to the market.

In contrast, carbon trigger funds and match funding (Table 3) provide a much stronger leveraging of private finance, directly stimulated by public investment. In the case of trigger funds, a proportion of public funding for projects is held back, and only released if a certain level of private investment can be secured within a particular time frame (typically after a project start date). The likelihood of securing private funding is then one of the selection criteria

**Table 3. Mechanisms for integrating public and private peatland payments for ecosystem services, based on focus group discussions.**

| Description | Strengths | Weaknesses |
|---|---|---|
| **1. Landscape-scale integration** | | |
| This is an organisational task; to enable public and private funding mechanisms to interact. It means overcoming mismatches in organisation scales, timelines, terminology, definitions, and metrics. Integration could happen in various ways but is scale dependent; a funding synergy in East Anglia won't be the same as one in Cumbria. Our recommendation is that public funding shapes itself around emerging private sector markets. | • Offers a single mechanism with options for both public and private finance and so is simple for land managers<br>• Can help identify trade-offs between ecosystem services<br>• Risks are shared between multiple private sector and public investors.<br>• A place-based approach adapted to local contexts and priorities with potential to feed into regional economic and community development | • Depending on the level of public funding integration, it could increase bureaucracy, and reduce the agility of private sector delivery<br>• Difficulty of attributing outcomes to funders may present additionality issues for funding via carbon codes, and there may be challenges in terms of WTO rules on what public funding can pay for in agriculture<br>• Operates at landscape scales, and so needs to be replicated and adapted for each new landscape, making scaling more challenging |
| **2. Funds delineation** | | |
| Locations or application windows are reserved for private and public funding, making room for the market to fund as many projects as possible before gap-filling with public funding where there would be market failure | • Clear 'lines of sight' between sources of funding and outcomes, helps with transparency.<br>• Helps boost scale and viability of projects.<br>• Funds multifunctionality. | • May not realise the full potential for 'leverage' presented by more fully integrated payments and action.<br>• Potential for funds to be mis-allocated–for example funding public access infrastructure that realistically will only be used for site management. |
| **3. Trigger funds** | | |
| 'Trigger funds' are government funds (directed at carbon, and / or other site outcomes) that would only be released once a certain level of private payments was reached. A single universal percentage level could be used, or stepped trigger levels could be used based on site prioritisation (ideally determined regionally) | • Allows Governments to co-fund ecosystem functions, without 'squeezing out' private sector finance.<br>• The effect of private finance triggering public funds could assist in demonstrating additionality. | • Set too low, trigger levels may have the effect of capping the level of private sector funding.<br>• Trigger funds would create organisational complexity |
| **4. Fund-matching as a default principle** | | |
| An extension of 'trigger funds' in that it establishes a wider default that public funds should only be issued on the basis that a level of private sector funds are already in place for a package of nature-based solutions. | • 'Signalling' to build confidence within the marketplace–avoiding both demand and supply side players being caught in an 'opportunity cost dilemma'. | • Risk that public-benefit oriented projects, where there is little private sector demand, will be disadvantaged. |
| **5. Targeting public sector funds via cost-benefit matrices** | | |
| Public funds would be adjusted according to a matrix of public benefit versus private finance potential. Stepped, or differential, rates of funding would need to be guided by a transparent set of tests, paying more for important public benefits where there is limited private finance potential | • Creates 'smarter' funding, 'stepping up' funds for more difficult, or public-good oriented schemes or locations.<br>• Provides a 'safety net' to fund valuable projects for which there is no private market | • Adds complexity, and requires a defensible and widely applicable set of tests. |
| **6. Carbon floor price guarantee mechanisms** (Fig 3) | | |
| Public funds can be used to provide guarantee mechanisms for PES markets that can help de-risk projects and funds for private investors. For efficient use of public funds, guarantees can be awarded via reverse auction mechanisms to allow projects or funds to compete with each other, thus optimising value for money. Guarantees effectively act as an option to sell carbon units in the future for project developers and/or funds, if the market cannot offer a more attractive price. This certainty over future income streams can unlock impact investment in addition to carbon finance, and incentivises developers to put forward projects because they are able to retain carbon units for sale at higher prices than they can achieve by pre-selling pending issuance units prior to verification. | • Avoids risk of crowding out private sector as it provides a potential revenue stream rather than just capital<br>• Value for money can be achieved through reverse auction mechanisms<br>• Criteria for auctions can be used to direct support into targeted subsectors and regions<br>• Ultimately if markets offer better prices, the guarantees may not be exercised thus freeing up public funds<br>• Proven to be effective in unlocking private capital in the UK in renewable energy and woodland creation markets<br>• Opportunity to create a profit capture mechanism to capture a proportion of market upside performance to recover capital for the public sector. | • Requires long-term public-sector commitments<br>• Does not explicitly deal with supply chain issues. While growing the market will help supply chains to develop, they may still require additional public support. |

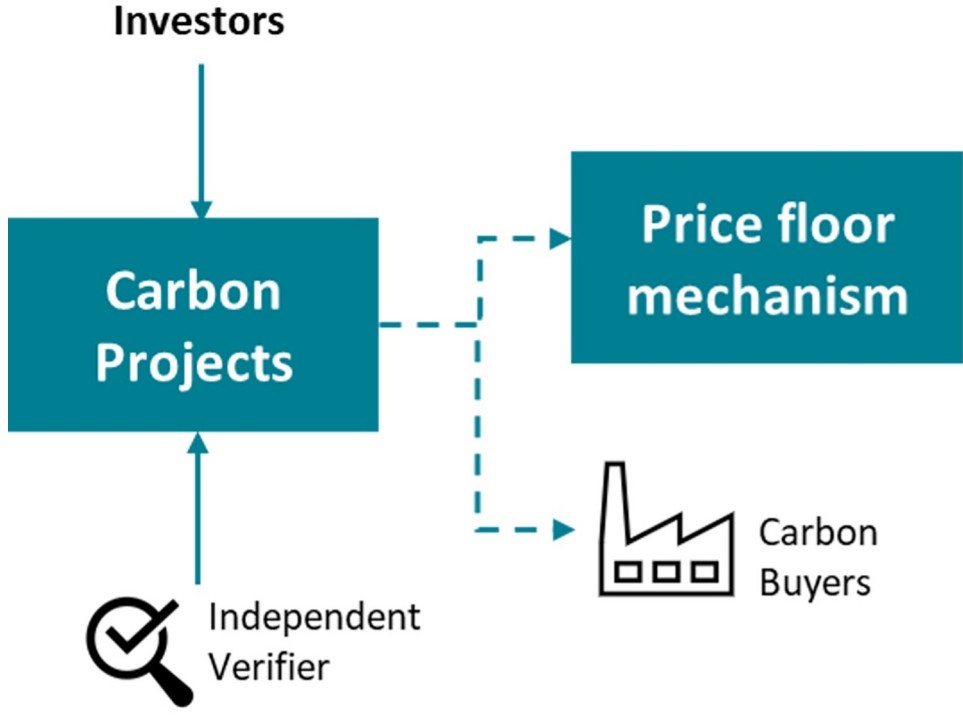

**Fig 3. A carbon guarantee mechanism provides projects with a floor price for carbon, reached via reverse auction and guaranteed by government, which can be triggered if projects are unable to find higher prices via carbon markets.**

for public funding of these projects, ranging from signed contracts and letters of intent to the plans and track record of the project developer or an intermediary they are working with to bring in private investment. Carbon trigger funds are complex to administer, and a proportion of projects won't ever actually get the private funding they seek. These projects won't draw down the second instalment of their grant, leaving the public funder with projects that don't reach their public good potential or private finance leveraging. However, if designed appropriately (e.g. restoring or planting up part of a site, rather than doing ground work across a whole site ready to investment that never materialises), projects may be able to deliver some benefits with public funding if they fail to get private investment. Because these are more likely to be in sites that deliver cost-effective climate benefits (to attract private investors), running a carbon trigger fund to stimulate private investment in the most attractive sites alongside funds delineation may provide governments with an attractive combination of leveraging power whilst considering issues of equity and distribution of benefits.

One of the challenges of funds delineation is identifying which sites should be prioritised for public versus private funding, and carbon trigger funds are likely to prioritise projects on the basis of their ability to leverage private funding, rather than the efficiency with which they can deliver carbon and other benefits. The carbon guarantee (Table 3; Fig 3) is more likely to identify the most cost-effective sites for private buyers because it relies on a reverse auction to prioritise sites to be supported by the guarantee. It has the potential to mobilise private capital to finance restoration in the long term, replacing the year-to-year public grant system, whilst giving confidence to both project developers and investors. The mechanism has so far been tested through the Woodland Carbon Guarantee in England, but it has the potential to be replicated in other ecosystem markets especially in national carbon markets or where

independent standards exist, such as the four peatland standards reviewed in Table 1. While the Woodland Carbon Guarantee typically relied on public funding to subsidise tree planting, future guarantee mechanisms could raise private capital to pay for capital works. For projects that require significant funding up front, for example to plant trees/hedgerows or do restoration works, there are currently two main options (which may be used in separately or combination). First, projects can use public grant funding, requiring the majority of project costs to be paid by the public and limiting the capacity for private funding to be leveraged. Second, some schemes allow projects to forward-sell "pending issuance units" once projects have been validated, prior to verification, at a significantly lower price than they would expect to achieve for verified units at a later date, to cover up-front project costs. However, this is a major disincentive for project developers who receive a fixed price for their carbon up front, which could have been worth significantly more had they been able to retain the units for future sale. However, the carbon guarantee mechanism opens an alternative funding mechanism for paying up-front project costs. If financing is agreed with impact investors with the terms made known to project developers prior to a reverse auction, these repayments (with interest) can be incorporated into bids, creating a floor price that enables projects to repay their finance, in addition to covering their own costs and profit. Investors may even provide commitments contingent on successfully accessing a guarantee. This then means that the carbon guarantee mechanism leverages both carbon finance and impact investment, giving the private sector confidence to invest in projects, knowing that project developers will be able to repay investment with a return by selling carbon units at the floor price via the guarantee if carbon markets are not able to sustain higher prices. If carbon markets are able to pay more than the floor price, then investors are repaid, and project developers retain any additional profits. There is a possibility under certain conditions that public funds (reserved in case the guarantee mechanism is triggered) are never used, and ecosystem services are delivered entirely via private funding, enabling public funding to be re-invested in future rounds. Conventionally, guarantees are offered at a project level, as shown in Fig 3.

Finally, it is possible to envisage the blending of public finance with multiple, co-ordinated private schemes. Funds delineation might reserve specific landscapes for private investment using a cost-benefit matrix, with carbon trigger funds, a match funding principle or guarantee mechanisms to leverage carbon finance, to pay for woodland creation, peatland and saltmarsh restoration or regenerative agriculture interventions that sequester and store soil carbon. Where interventions are too expensive to be paid via carbon finance alone, payments for biodiversity might be layered on top of the carbon finance to make projects financially feasible. If co-ordinated at a landscape scale by an entity such as LENs, it may be possible to aggregate demand for layered, fungible services such as carbon and biodiversity, with non-fungible services such as water quality and animal welfare (defined as a public good in UK law) for buyers seeking to reduce risks to their business from climate change or other drivers of change. At the same time, some land within the same landscape may be eligible for entry to agri-environment schemes to pay for interventions that deliver services not included in private schemes.

## 6 Conclusions

This paper has provided an empirical basis for a new typology of ecosystem markets based on schemes currently operating or under development in the UK and in European peatlands. Each have distinct operational scales of investment and delivery, modes of funding and governance models. Of particular interest are emerging regional ecosystem markets, which are stimulating and meeting demand for ecosystem services by framing demand in relation to business risks and aggregating both demand side interests and the supply of services, overcoming free-

rider effects and minimising trade-offs between ecosystem services across a landscape. Contrary to assumptions underpinning traditional PES schemes, taking this approach may lead to strong and resilient demand for ecosystem services in the absence of tight coupling between payments and provision of benefits. However, integration of these regional ecosystem markets with national carbon markets and green finance mechanisms may provide an expanded range of investors and land uses from which a much wider range of services can be provided.

The integration of private schemes may also make it possible to co-ordinate more effectively with public funding for ecosystem services, prioritising public funding towards landscapes and services not paid for by the market, and increasing the diversity and amount of funding for sustainable land management interventions. While the options for integrating private ecosystem markets proposed here are currently theoretical, there are now attempts to apply these integrative governance models in practice. Achieving integration between schemes is increasingly important as private ecosystem markets proliferate around the world. However, as separate schemes proliferate, so does the likelihood of competition and trade-offs between services provided by different schemes. The need to manage these trade-offs and ensure private investment contributes to multifunctional landscapes is therefore a key driver for considering how schemes can more effectively integrate with each other.

As publicly funded schemes also become more PES-like in many countries, there is a risk of perverse outcomes if public funding pays for services that would otherwise have been provided by the market. However, by designing future public schemes to complement private ecosystem services, it may be possible to avoid these markets being crowded out, and even use public funding to leverage private investment, for example via carbon trigger funds (Table 3). As Government budgets come under increasing pressure, stimulating ecosystem markets could help fill the funding gap, contributing to a green post-COVID economic recovery, and increasing the likelihood that ambitious climate and biodiversity targets are met. However, to unlock this private finance, mechanisms need to be developed to ensure public and private funding can be successfully blended in future nature-based projects, for example integrating funds delineation with carbon trigger funds or carbon guarantees (see discussion of Table 3). Robust standards (akin to those developed for peatland restoration in Europe) are needed to govern the development of new markets in a wider range of land uses and habitats, to provide investor confidence and ensure outcomes are delivered. Public funding may also be used to help these new markets develop investment pipelines with projects that are ready for investment with the associated staff and governance mechanisms to channel investment scale capital into nature-based solutions. In some contexts regulation may be considered, for example the integration of Net Biodiversity Gain in the planning system, requiring developers to make (typically offsite) provisions to compensate for biodiversity losses and provide additional biodiversity gains. Government funding could also help unlock supply by employing facilitators to explain opportunities to owners and managers of land and marine assets, simplifying and democratising access to private finance.

In conclusion, much still needs to be done to stimulate and integrate ecosystem markets, but with the right support and design, it may be possible to integrate multiple sources of private investment with public funding to start delivering the levels of funding needed to address the twin challenges of climate change and biodiversity loss. The typology developed in this paper was developed with reference to the analysis of private ecosystem markets in the UK and Europe, so care should be taken in applying this more widely. However, as the number of different markets and financial instruments facilitating the sale of ecosystem services grows, the conceptual clarity brought by this typology may aid those seeking to develop or access private finance in the context of green recovery and meeting net zero and other targets.

## Supporting information

**S1 File.**
(DOCX)

## Acknowledgments

There are no patents, products in development or marketed products associated with this research to declare. Thanks to Dr Gordon Mitchell (University of Leeds) for input to one of the reports that formed an early basis for this paper. Thanks to Julie Martin-Ortega (University of Leeds), Kerry Waylen (James Hutton Institute), Bryan Adkins (United Nations Environment Programme), Jo Pike (Scottish Wildlife Trust) and Rosmarie Katrin Neumann (Newcastle University) for useful comments on an earlier draft of this paper.

## Author Contributions

**Conceptualization:** Mark S. Reed, Tom Curtis, Arjan Gosal, Guy Ziv, Richard G. Fitton, Alicia C. Gibson, Alex C. Hume, Jamie L. Mansfield.

**Data curation:** Mark S. Reed, Arjan Gosal, Helen Kendall, Asger Strange Olesen.

**Formal analysis:** Mark S. Reed, Arjan Gosal, Helen Kendall, Sarah Pyndt Andersen, Asger Strange Olesen.

**Funding acquisition:** Mark S. Reed, Tom Curtis, Guy Ziv, Asger Strange Olesen.

**Investigation:** Mark S. Reed, Arjan Gosal, Helen Kendall, Asger Strange Olesen.

**Methodology:** Mark S. Reed, Arjan Gosal, Sarah Pyndt Andersen, Guy Ziv, Asger Strange Olesen.

**Project administration:** Mark S. Reed, Arjan Gosal, Asger Strange Olesen.

**Resources:** Mark S. Reed, Arjan Gosal.

**Supervision:** Mark S. Reed, Guy Ziv, Asger Strange Olesen.

**Writing – original draft:** Mark S. Reed, Tom Curtis, Arjan Gosal, Helen Kendall, Sarah Pyndt Andersen, Guy Ziv, Matthew Hay, Simone Martino, Asger Strange Olesen, Stephen Prior, Christopher Rodgers.

**Writing – review & editing:** Mark S. Reed, Tom Curtis, Arjan Gosal, Helen Kendall, Guy Ziv, Anais Attlee, Richard G. Fitton, Alicia C. Gibson, Alex C. Hume, David Hill, Jamie L. Mansfield, Asger Strange Olesen, Christopher Rodgers, Hannah Rudman, Franziska Tanneberger.

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
