## [Decision Letter · Decision Letter 0]

2 Mar 2021

PONE-D-20-40133

Integrating ecosystem markets to co-ordinate landscape-scale public benefits from nature

PLOS ONE

Dear Dr. Reed,

Thank you for submitting your manuscript to PLOS ONE. After careful consideration, we feel that it has merit but does not fully meet PLOS ONE’s publication criteria as it currently stands. Therefore, we invite you to submit a revised version of the manuscript that addresses the points raised during the review process.

We look forward to receiving your revised manuscript.

Kind regards,

Neville Crossman, Ph.D.

Academic Editor

PLOS ONE

Journal Requirements:

Additional Editor Comments (if provided):

I encourage the authors to respond very carefully to both reviews and put the effort into the major revision.

"Mark Reed is Research Lead for IUCN UK Peatland Programme and sits on the Executive Board for the Peatland Code. Tom Curtis is a founding partner of 3Keel and helped develop Landscape Enterprise Networks. Matthew Hay is a Project Manager and Stephen Prior is co-founder of Forest Carbon Ltd. David Hill is founding owner of Environment Bank Ltd."

Reviewers' comments:

Reviewer's Responses to Questions

**Comments to the Author**

1. Is the manuscript technically sound, and do the data support the conclusions?

Reviewer #1: Partly

Reviewer #2: Yes

2. Has the statistical analysis been performed appropriately and rigorously? 

Reviewer #1: N/A

Reviewer #2: N/A

3. Have the authors made all data underlying the findings in their manuscript fully available?

Reviewer #1: Yes

Reviewer #2: Yes

4. Is the manuscript presented in an intelligible fashion and written in standard English?

Reviewer #1: Yes

Reviewer #2: Yes

5. Review Comments to the Author

Reviewer #1: This paper presents an attempt to identify the possibilities and impediments to integrated ecosystem service markets through an exploration of the emergence and framing of a small set of UK ‘ecosystem services’ markets and EU peatland markets. There are some clear lessons from the evaluation and the attempt to explore unification options has promise. Yet the markets explored are limited geographically and institutionally, and the exploration itself has some substantial weaknesses that I feel need to be addressed to support the conclusions drawn. For these reasons, expanded below, I have recommended reject.

Major comments:

It is not clear what is considered a ‘market’ and therefore considered in-scope or out of scope and why. Although there are several discussions in the paper it does not seem to me that there is clarity around supply and demand issues. At first glance, particularly from the discussion in the introduction, it appears only two-sided markets consisting of private sellers and buyers are within scope. Later in the paper it becomes clear that there are a range of different ‘sellers’, some of which could not be considered private, whilst some ‘markets’ may have large components of public funding (i.e. public buyers). The inclusion of investment funds and bonds is also complicating, on the one hand they appear to represent a private ‘demand’ in the market, however since in at least some instances the finance must be repaid at a future point in time they seem to be more about credit supply and risk management in environmental markets rather than actual market participants on either the demand or supply side. This confusion about what the paper covers is reflected in the commentary in section 2 ‘debate emerged over whether the analysis was considering schemes, markets, or stakeholder engagement frameworks’. In a nutshell the paper requires greater clarity about the ‘market’ analysed.

This lack of clarity around what is considered a market means that it is unclear whether schemes such as The Countryside Stewardship Scheme should be included (publicly designed and funded agri-environment scheme), or some activities of not for profit nature conservation groups such as Wildlife Trusts (e.g. purchase and management of land for ecosystem services). More importantly, limiting the geographic scope of the paper to the UK and three markets focusing on peatland in Europe means that the potential for generalisation across landscape scale public good markets is limited. For example, the not-for-profit sector plays a much larger role in market formation and facilitation in the US (see for example The Nature Conservancy facilitates a number of landscape scale initiatives inclusive of markets) where there are a large number of different initiatives seeking landscape scale public good outcomes (at least in part) using markets. With respect to this point – the introduction should be altered to clarify that the exploration is limited to ‘ecosystem markets currently in operation or close to market in the UK, and peatland markets in Germany, Switzerland and the Netherlands’.

Exploration of supply and demand issues (e.g. as specified in row in Table 1) only very briefly explores prices (which the discussion in text makes clear are not market clearing prices which would emerge were supply and demand to intersect) and does not address broader supply and demand issues (with the exception of BIF – where the discussion simply states ‘no problem expected’). A paper wishing to explore landscape scale outcomes should in my view credibly identify in the relevant case study markets whether there are constraints from both a supply and demand perspective. This step would help to identify overall potential feasibility, particularly given that some landscape scale outcomes can only be achieved with substantial participation – i.e. supply of the ecosystem service. Furthermore, the results conclude “… engagement with suppliers (typically landowners and managers) was a challenge for all schemes …” but this appears in conflict with the conclusion in the data summary in Table 1 where no such issues were identified. Indeed, it would be useful to identify whether these schemes (or an integrated version of them) are able to adequately address the landscape scale problem or the supply and demand is inconsistent with the solution scale required.

The inclusion of national carbon markets into the discussion section is problematic from a language and practicality perspective. First, voluntary and compliance carbon markets were specifically excluded in section 2 yet seem to re-appear in Table 2 and in the discussion of types of ecosystem markets (because several case study schemes are categorized as such). Second, the basis for the conclusions in Table 2 and Section 4.1 are unclear. Early references to carbon markets (section 2) refer only to voluntary carbon markets. Noting the examples given in Table 2, the authors are indicating that there is a class of markets that are largely driven by carbon and operate anywhere within the boundaries of the relevant country. This should be made much more clear given the earlier language in the paper. Furthermore, there will likely be interaction with formal carbon markets which in effect set a floor price for this ecosystem service.

The discussion of regional ecosystem markets would benefit from reference to the economic theory around club goods which these types of investment coalitions appear to be consistent with. This appears relevant at several points in sections 4 and 5.

The paper presents a number of novel points which could be emphasised to further strengthen the paper:

• There is a unfortunately a conflict between pure public goods represented by carbon and (at least partly) spatially delineated public goods such as flood mitigation – so integration of regional markets could either strengthen pathways to carbon markets and therefore the market OR weaken regional markets via the potential for institutional rules to prevent or complicate ecosystem service ‘stacking’. I suggest more attention could be paid to this point.

• What are the challenges in moving from the facilitated, network style markets brokered by individuals towards a more anonymised trading approach? Is an anonymised trading approach incompatible with the market structures explored in the paper? (or are some of these markets essentially boutique?)

• Regional ecosystem markets are essentially parallel markets under any scenario in the paper (because they are geographically focused). This means that they would always remain separate – but what can be learnt across them that facilitates integration in other ways across the overlapping national or ecosystem specific markets and green finance schemes?

• Robust standards appear to make goods more ‘fungible’. Is there a role or a pathway that moves ecosystem goods along a continuum towards fungibility and market formation in the way that some of these markets are developing? It is also important to note that the ‘fungibility’ of carbon is due to agreed conversion of the green house gas forcing of a range of gases emitted to the atmosphere (e.g. methane particularly in a landscape scale setting).

Reviewer #2: Review of: Integrating ecosystem markets to co-ordinate landscape-scale public benefits from nature

General:

The manuscript is: 1) a review of several PES of varying types in UK and Europe, 2) an attempt to develop a typology of PES schemes, 3) an evaluation of advantages, disadvantages of different type of PES, and 4) evaluation of potential to further integrate ES markets.

In general, I think that the topic is important in the context of a growing and diverse array of mechanisms to focus on ES service outcomes of investment and payments directly for ES outcomes and relatively little literature that looks at and evaluates of the diversity in this growing sector. I think that the methods applied interviews, focus groups and secondary research are appropriate for the task at hand. I also believe that a refined and improved version would provide valuable insight and be read, valued and cited by the relevant research and policy community.

I do think that there are a number of writing and organisational issues in manuscript that would require improvement prior to the manuscript being published.

I also feel that some very general conclusions seem pretty overstated. For example the manuscript ends by concluding that

“with the right support and design, it may be possible to

integrate multiple sources of private investment with public funding to deliver the levels of funding needed to address the twin challenges of climate change and biodiversity loss.”

some additional specific comments are embedded in the attached commented original manuscript

A more realistic conclusion that is backed by the evidence presented is that: we are currently seeing emergence of discussion and early small scale “niche market” implementation around the right support and design to integrate m.ultiple sources of private investment with public funding, but we remain a long way from being able to deliver the levels of funding needed to truly adequately address the twin challenges of climate change and biodiversity loss.”

Specific

Intro – the first part of discussion in section 4 – highlighted in yellow for the authors – is introductory material – not discussion in the sense that it is general and speaks to why the topic treated is an issue – its not discussion of the specific study findings.

Move this text and using in re-write of intro.

Methods- I’m happy with this section. It provides brief but complete explanation of what was done and I think that the appropriate and adequate methods are chosen and adequately explained.

Results- The table in this section is a useful way to summarise a lot of relevant information and analysis. I read it carefully and referred to when reading discussion. The one thing that I did wonder about as the lumping together of what are two quite different public regulation driven markets – carbon and biodiversity offset. The former is easier to measure with better developed standardised protocols. The challenges of equivalence has made the latter more challenging to date.

I was less pleased with the page and a half of results reporting that followed in two blocks of dot pointed text. I felt like that information could be better organised into a set of subheaded sections and that in some cases themes re-emerged under different dot points rather than being discussed all at once. Not sure what exactly the right sub-heading would be but could include: source of demand public versus private, verification/validation, intermediaries and supply/demand coordination, co-benefits, additionality.

Discussion

Mostly good discussion – I did wonder about extent to which conclusions were specific to the particular schemes investigated as opposed to the class of schemes generally. For example an advantage of local scheme described

“successful aggregation of demand was in part due to the proactive role of trusted business-to-business brokers, compared to the national carbon markets, which tended to be managed by organisations with very different cultures and language (typically Non-Governmental

Organisations, research institutes or Government agencies), who played a more passive role in engaging with investors.”

I’m also aware that some green investors like Kilter Ltd investing for the Victorian Superannuation uses a similar model. Is this a chacteristic of local versus green investor?

Conclusion We are still talking about a scale that is small and niche, this doesn’t really come across in conclusions. The conclusions can also be a bit better articulated regarding what recommendations for role of government.

6. PLOS authors have the option to publish the peer review history of their article (what does this mean?). If published, this will include your full peer review and any attached files.

Reviewer #1: No

Reviewer #2: **Yes: **jeffery d connor

---

## [Author Response · Author response to Decision Letter 0]

21 May 2021

Reviewer #1

Reviewer: This paper presents an attempt to identify the possibilities and impediments to integrated ecosystem service markets through an exploration of the emergence and framing of a small set of UK ‘ecosystem services’ markets and EU peatland markets. There are some clear lessons from the evaluation and the attempt to explore unification options has promise. Yet the markets explored are limited geographically and institutionally, and the exploration itself has some substantial weaknesses that I feel need to be addressed to support the conclusions drawn. For these reasons, expanded below, I have recommended reject.

Major comments:

It is not clear what is considered a ‘market’ and therefore considered in-scope or out of scope and why. Although there are several discussions in the paper it does not seem to me that there is clarity around supply and demand issues. At first glance, particularly from the discussion in the introduction, it appears only two-sided markets consisting of private sellers and buyers are within scope. Later in the paper it becomes clear that there are a range of different ‘sellers’, some of which could not be considered private, whilst some ‘markets’ may have large components of public funding (i.e. public buyers). The inclusion of investment funds and bonds is also complicating, on the one hand they appear to represent a private ‘demand’ in the market, however since in at least some instances the finance must be repaid at a future point in time they seem to be more about credit supply and risk management in environmental markets rather than actual market participants on either the demand or supply side. This confusion about what the paper covers is reflected in the commentary in section 2 ‘debate emerged over whether the analysis was considering schemes, markets, or stakeholder engagement frameworks’. In a nutshell the paper requires greater clarity about the ‘market’ analysed.

Response:

• We have now provided a definition for ecosystem markets in the first paragraph of the methods section: “For the purposes of our sample, we defined ecosystem markets as full developed platforms that could facilitate ongoing exchanges between multiple private buyers and sellers of ecosystem services in the UK and in European peatlands.”

• We have listed the inclusion and exclusion criteria for our sample more explicitly in this paragraph too, addressing each of the points helpfully made by the reviewer: “As long as the scheme was designed primarily to facilitate private investment (and this was required in additionality criteria), schemes that also leveraged public investment were included in the sample. Schemes could facilitate investment directly through the purchase of ecosystem services or indirectly by providing credit supply and risk management, as long as the goal of the financial mechanism was to facilitate investment in ecosystem services.”

• The point about including investment funds and bonds as “buyers” is correct – they are not technically buying ecosystem services, rather facilitating their production. Our point however, is that without that private finance, the services would not be supplied, so these are included in our definition of markets that “facilitate” buying and selling of ecosystem services. Green finance is a major player in the ecosystem market landscape, and to exclude this based on a narrow definition of markets would significantly constrain the utility of the paper for readers in policy and practice, where a growing number of finance mechanisms are making new ecosystem markets possible.

• The final point about debate (not “confusion”) over which markets to include during the scoping phase has been clarified with reference to the clearer articulation of exclusion and inclusion criteria now included in the introduction to the methods section.

Reviewer: This lack of clarity around what is considered a market means that it is unclear whether schemes such as The Countryside Stewardship Scheme should be included (publicly designed and funded agri-environment scheme), or some activities of not for profit nature conservation groups such as Wildlife Trusts (e.g. purchase and management of land for ecosystem services). More importantly, limiting the geographic scope of the paper to the UK and three markets focusing on peatland in Europe means that the potential for generalisation across landscape scale public good markets is limited. For example, the not-for-profit sector plays a much larger role in market formation and facilitation in the US (see for example The Nature Conservancy facilitates a number of landscape scale initiatives inclusive of markets) where there are a large number of different initiatives seeking landscape scale public good outcomes (at least in part) using markets. With respect to this point – the introduction should be altered to clarify that the exploration is limited to ‘ecosystem markets currently in operation or close to market in the UK, and peatland markets in Germany, Switzerland and the Netherlands’.

Response:

• Publicly funded schemes were excluded from the study on the basis that the scope was private ecosystem markets. The definition of markets and scope including more explicit inclusion and exclusion criteria has now been clarified, and we have explicitly stated in reference to our definition that publicly funded schemes that did not require private finance for their operation were excluded from the study

• We agree that the potential for generalisation is limited by the geographical scope of our research. In the first sentence of the conclusion, we already made it clear that “This paper has provided an empirical basis for a new typology of ecosystem markets based on schemes currently operating or under development in the UK and in European peatlands”. However, we have made this more explicit now, at the end of the conclusion, in response to this helpful comment by the reviewer. The introduction already states the scope clearly when the aim of the paper is introduced: “This paper therefore uses a comparative analysis of existing private ecosystem markets in operation or close to market at national and sub-national scales in the UK and elsewhere in Europe”. Moreover, this is further clarified in the first objective of the paper under this broad aim, to Develop a typology of ecosystem markets by comparing ecosystem markets currently in operation or close to market in the UK, Germany, Switzerland and the Netherlands”.

Reviewer: Exploration of supply and demand issues (e.g. as specified in row in Table 1) only very briefly explores prices (which the discussion in text makes clear are not market clearing prices which would emerge were supply and demand to intersect) and does not address broader supply and demand issues (with the exception of BIF – where the discussion simply states ‘no problem expected’). A paper wishing to explore landscape scale outcomes should in my view credibly identify in the relevant case study markets whether there are constraints from both a supply and demand perspective. This step would help to identify overall potential feasibility, particularly given that some landscape scale outcomes can only be achieved with substantial participation – i.e. supply of the ecosystem service. Furthermore, the results conclude “… engagement with suppliers (typically landowners and managers) was a challenge for all schemes …” but this appears in conflict with the conclusion in the data summary in Table 1 where no such issues were identified. Indeed, it would be useful to identify whether these schemes (or an integrated version of them) are able to adequately address the landscape scale problem or the supply and demand is inconsistent with the solution scale required.

Response:

• The reviewer refers specifically to Table 1, which provides only a high-level summary of supply and demand issues. However, there is an entire section on supply and demand issues in supplementary material. We wanted to discuss each row of the table in depth, whilst keeping the main body of the paper as concise as possible for readers, and so put the additional results and discussion in supplementary material. Given that this is easily missed, in addition to signposting this in the main text, we have also linked to supplementary material in the legend of Table 1, and we have signposted the supplementary material at each of the other points Table 1 is referenced in the main article. 

• Table 1 has now been extended to summarise issues with supply that are discussed in supplementary material to avoid the apparent contradiction between the main article and this table. The conclusion itself (in the first bullet list in section 3) has also been amended to better reflect the summary as it now stands in Table 1.

• In addition to the extensive discussion of supply and demand issues in supplementary material, we have now discussed pricing more explicitly in the main article, considering which schemes reached clearing prices via negotiation between buyers and sellers versus schemes that based prices on the cost of interventions (first bullet list in section 3). We have also discussed stacking versus bundling of payments and how these payments could meet additionality criteria across multiple schemes (second bullet list in section 3). 

Reviewer: The inclusion of national carbon markets into the discussion section is problematic from a language and practicality perspective. First, voluntary and compliance carbon markets were specifically excluded in section 2 yet seem to re-appear in Table 2 and in the discussion of types of ecosystem markets (because several case study schemes are categorized as such). Second, the basis for the conclusions in Table 2 and Section 4.1 are unclear. Early references to carbon markets (section 2) refer only to voluntary carbon markets. Noting the examples given in Table 2, the authors are indicating that there is a class of markets that are largely driven by carbon and operate anywhere within the boundaries of the relevant country. This should be made much more clear given the earlier language in the paper. Furthermore, there will likely be interaction with formal carbon markets which in effect set a floor price for this ecosystem service.

Response:

• Although the exclusion of international voluntary carbon markets was already justified in the introduction and methods, this has been made more explicit by listing this as an example of a market that was excluded from the study in the revised introduction to the methods section

• Table 2 did not and still does not mention voluntary or compliance markets at any point. We have however clarified our definition of national carbon markets and explained more clearly how these differ from voluntary and compliance markets now in the definition of carbon markets in the text (section 4.1). As part of this, we have also explained the very loose link to carbon market rates, given that prices tended to be based primarily on project costs 

• We have introduced the idea of carbon guarantees with set floor prices negotiated via revers auctions, which may be used to provide liquidity and confidence within national carbon markets in the discussion (see also the additional row in Table 3 and Figure 3)

Reviewer: The discussion of regional ecosystem markets would benefit from reference to the economic theory around club goods which these types of investment coalitions appear to be consistent with. This appears relevant at several points in sections 4 and 5.

Response: The paper now contains a proposal on how regional ecosystem markets can be interpreted in terms of “club goods”. The property of rivalry is interpreted as a form of “congestion” that generates disutility in investors (facilitating free-riding) if the number of investors is too high compared to the environmental benefit received. Although other investments may reduce the cost of participation in the schemes (especially for fixed cost schemes), they may induce free-riding if the marginal benefits perceived by the provision of ecosystem services occurs at a lower rate than cost reduction. A mechanism with fixed and variable cost is proposed, with government paying up front fixed cost of the investment, and private buyers negotiating prices of services with suppliers up to the point that the incremental change of participating to the markets equalise the benefits received. Due to space constraints, this has been integrated with supplementary material at the end of the paper.

Reviewer: The paper presents a number of novel points which could be emphasised to further strengthen the paper:

• There is a unfortunately a conflict between pure public goods represented by carbon and (at least partly) spatially delineated public goods such as flood mitigation – so integration of regional markets could either strengthen pathways to carbon markets and therefore the market OR weaken regional markets via the potential for institutional rules to prevent or complicate ecosystem service ‘stacking’. I suggest more attention could be paid to this point.

• What are the challenges in moving from the facilitated, network style markets brokered by individuals towards a more anonymised trading approach? Is an anonymised trading approach incompatible with the market structures explored in the paper? (or are some of these markets essentially boutique?)

• Regional ecosystem markets are essentially parallel markets under any scenario in the paper (because they are geographically focused). This means that they would always remain separate – but what can be learnt across them that facilitates integration in other ways across the overlapping national or ecosystem specific markets and green finance schemes?

• Robust standards appear to make goods more ‘fungible’. Is there a role or a pathway that moves ecosystem goods along a continuum towards fungibility and market formation in the way that some of these markets are developing? It is also important to note that the ‘fungibility’ of carbon is due to agreed conversion of the green house gas forcing of a range of gases emitted to the atmosphere (e.g. methane particularly in a landscape scale setting).

Response:

• These are all interesting and pertinent questions, and we have done our best to answer these in the paper. However, we have already extended the paper significantly and to do these questions justice would take significantly more room than we have available given the word limit. As a result, some of these questions remain unanswered in this paper and we hope to explore them in greater depth in future work. 

• Nevertheless, we have integrated a more detailed discussion of bundling versus stacking of ecosystem services into section 3 now

• Although technically separate, we explain how regional ecosystem markets and national carbon markets can interact in more or less integrative ways in the discussion

Reviewer #2

Reviewer: General: The manuscript is: 1) a review of several PES of varying types in UK and Europe, 2) an attempt to develop a typology of PES schemes, 3) an evaluation of advantages, disadvantages of different type of PES, and 4) evaluation of potential to further integrate ES markets. In general, I think that the topic is important in the context of a growing and diverse array of mechanisms to focus on ES service outcomes of investment and payments directly for ES outcomes and relatively little literature that looks at and evaluates of the diversity in this growing sector. I think that the methods applied interviews, focus groups and secondary research are appropriate for the task at hand. I also believe that a refined and improved version would provide valuable insight and be read, valued and cited by the relevant research and policy community.

I do think that there are a number of writing and organisational issues in manuscript that would require improvement prior to the manuscript being published. I also feel that some very general conclusions seem pretty overstated. For example the manuscript ends by concluding that: “with the right support and design, it may be possible to integrate multiple sources of private investment with public funding to deliver the levels of funding needed to address the twin challenges of climate change and biodiversity loss.”

… A more realistic conclusion that is backed by the evidence presented is that: we are currently seeing emergence of discussion and early small scale “niche market” implementation around the right support and design to integrate m.ultiple sources of private investment with public funding, but we remain a long way from being able to deliver the levels of funding needed to truly adequately address the twin challenges of climate change and biodiversity loss.”

Response: This statement has been toned down and followed by a sentence unpacking some of the limitations of our work, cautioning over-generalisation. 

Reviewer: some additional specific comments are embedded in the attached commented original manuscript

Response: These have all been considered and where relevant actioned

Reviewer: Specific

Intro – the first part of discussion in section 4 – highlighted in yellow for the authors – is introductory material – not discussion in the sense that it is general and speaks to why the topic treated is an issue – its not discussion of the specific study findings. Move this text and using in re-write of intro.

Methods- I’m happy with this section. It provides brief but complete explanation of what was done and I think that the appropriate and adequate methods are chosen and adequately explained.

Results- The table in this section is a useful way to summarise a lot of relevant information and analysis. I read it carefully and referred to when reading discussion. The one thing that I did wonder about as the lumping together of what are two quite different public regulation driven markets – carbon and biodiversity offset. The former is easier to measure with better developed standardised protocols. The challenges of equivalence has made the latter more challenging to date.

I was less pleased with the page and a half of results reporting that followed in two blocks of dot pointed text. I felt like that information could be better organised into a set of subheaded sections and that in some cases themes re-emerged under different dot points rather than being discussed all at once. Not sure what exactly the right sub-heading would be but could include: source of demand public versus private, verification/validation, intermediaries and supply/demand coordination, co-benefits, additionality.

Response:

• The first part of the discussion has been moved to a new “background” section between the introduction and methods sections to avoid making the introduction too long and unfocussed

• The results were originally organised under sub-headings which matched the rows of Table 1, but this was significantly longer than the available word limit, so the extended results were moved to supplementary material, where these sub-headings and a more in-depth description of our findings can be found. As a comparative analysis, we grouped our high-level summary of the findings into differences and commonalities between the schemes (the two bullet lists). In some cases, there are key differences and commonalities within the same theme, which we now realise may appear repetitive. We have therefore made it clearer that the two groups of points are only a summary, and that readers should refer to Table 1 and supplementary material for more detailed information. We have also extended the point about the sale of ecosystem services in the second list (differences between schemes), as pricing is already covered in the first list (commonalities between schemes), to make it clearer that the key differences are around their approach to bundling versus stacking of payments for different ecosystem services

Reviewer: Mostly good discussion – I did wonder about extent to which conclusions were specific to the particular schemes investigated as opposed to the class of schemes generally. For example an advantage of local scheme described

“successful aggregation of demand was in part due to the proactive role of trusted business-to-business brokers, compared to the national carbon markets, which tended to be managed by organisations with very different cultures and language (typically Non-Governmental

Organisations, research institutes or Government agencies), who played a more passive role in engaging with investors.”

I’m also aware that some green investors like Kilter Ltd investing for the Victorian Superannuation uses a similar model. Is this a chacteristic of local versus green investor?

Response: We have checked this and other points made in the discussion, and they are clearly linked to the specific schemes in our results (in the case above, to LENs). However, as the scheme was only mentioned towards the top of the paragraph, we have mentioned it again later to make this clear and the quoted sentence now reads, “The successful aggregation of demand in LENs…”

Reviewer: Conclusion We are still talking about a scale that is small and niche, this doesn’t really come across in conclusions. The conclusions can also be a bit better articulated regarding what recommendations for role of government.

Response: Conclusions have been toned down and limitations added (see response above). We have added an extended discussion of the options for government to blend public and private finance, focusing on how three out of the six suggested mechanisms in Table 3 could be combined to use public funding to leverage private investment. These more detailed recommendations are also now highlighted in the conclusions.

---

## [Decision Letter · Decision Letter 1]

27 Sep 2021

Integrating ecosystem markets to co-ordinate landscape-scale public benefits from nature

PONE-D-20-40133R1

Dear Dr. Reed,

We’re pleased to inform you that your manuscript has been judged scientifically suitable for publication and will be formally accepted for publication once it meets all outstanding technical requirements.

Kind regards,

Neville Crossman, Ph.D.

Academic Editor

PLOS ONE

Additional Editor Comments (optional):

Reviewers' comments:

Reviewer's Responses to Questions

**Comments to the Author**

1. If the authors have adequately addressed your comments raised in a previous round of review and you feel that this manuscript is now acceptable for publication, you may indicate that here to bypass the “Comments to the Author” section, enter your conflict of interest statement in the “Confidential to Editor” section, and submit your "Accept" recommendation.

Reviewer #2: All comments have been addressed

2. Is the manuscript technically sound, and do the data support the conclusions?

Reviewer #2: Yes

3. Has the statistical analysis been performed appropriately and rigorously? 

Reviewer #2: N/A

4. Have the authors made all data underlying the findings in their manuscript fully available?

Reviewer #2: Yes

5. Is the manuscript presented in an intelligible fashion and written in standard English?

Reviewer #2: Yes

6. Review Comments to the Author

Reviewer #2: I'm happy with the revisions, the authors have addressed my initial concerns and those of the other reviewer well. This is a very good contribution to the literature done thoroughly with appropriate methods and well written - ready to publish in Plos Oen

7. PLOS authors have the option to publish the peer review history of their article (what does this mean?). If published, this will include your full peer review and any attached files.

Reviewer #2: **Yes: **jeffery d connor

---

## [Editor Report · Acceptance letter]

10 Dec 2021

PONE-D-20-40133R1 

Integrating ecosystem markets to co-ordinate landscape-scale public benefits from nature 

Dear Dr. Reed:

I'm pleased to inform you that your manuscript has been deemed suitable for publication in PLOS ONE. Congratulations! Your manuscript is now with our production department. 

Kind regards, 

on behalf of

Dr. Neville Crossman 

Academic Editor

PLOS ONE